# Waste-Based Adsorbents for the Removal of Phenoxyacetic Herbicides from Water: A Comprehensive Review

**Magdalena Blachnio [1],\* , Krzysztof Kusmierek [2] , Andrzej Swiatkowski [2] and Anna Derylo-Marczewska [1]**

1 Faculty of Chemistry, Maria Curie-Sklodowska University, M. Curie-Sklodowska Sq. 3, 20-031 Lublin, Poland; anna.derylo-marczewska@mail.umcs.pl

2 Institute of Chemistry, Military University of Technology, gen. S. Kaliskiego St. 2, 00-908 Warszawa, Poland; krzysztof.kusmierek@wat.edu.pl (K.K.); a.swiatkowski@wp.pl (A.S.)

\* Correspondence: magdalena.blachnio@mail.umcs.pl; Tel.: +48-8153-756-37

**Abstract:** An overview of different adsorbents, based on agricultural and household waste, for chlorophenoxy herbicides removal from water is presented. Several groups of adsorbents are discussed, taking into account the modification method used on the initial material and the type of final product obtained. The adsorbent characteristics and the conditions of the adsorption measurements are given, and a discussion on the obtained results is presented, along with a theoretical description, following the application of various equations and models. A group of the most effective adsorbents is indicated, based on the analysis of the adsorption capacity, towards 2,4-D and/or MCPA, and the adsorption rate. Some important problems connected with adsorbent utility are discussed, taking into account economic and ecological aspects. Moreover, the effectiveness of the analyzed materials is observed through the analysis of its interactions with other components present in real systems.

**Keywords:** adsorbents based on agricultural waste; adsorbents based on household waste; phenoxyacetic herbicides; herbicides removal; herbicides adsorption; herbicides adsorption kinetics

## 1. Introduction

Pesticides are a group of chemical compounds that are used to combat plant diseases, remove and destroy weeds, combat pests, and treat seeds. Over the past few decades, there has been a constant increase in plant and animal production around the globe, which has resulted in the growing demand for pesticides. At the global level, in the last decade, a 50% increase in pesticide use compared to the 1990s has been observed. In 2020, the total amount of pesticides used in agriculture was on the level of 2.7 million tons of active ingredients, which corresponds to approximately 7.2 million tons of formulated pesticide products. Taking into account the specific target groups, herbicides accounted for the largest share in the total consumption (52%), followed by fungicides (23%), insecticides (18%), and other pesticides (7%) [1].

The dynamic increase in the use of pesticides poses the risk of water pollution with such compounds. This problem is substantial in countries with a high share of agricultural land and those with low water resources, and in countries where surface water is the main source of drinking water for the inhabitants. According to recent data, on average in the whole of Europe, 37% of the public water supply comes from surface water [2], while in Bulgaria, Ireland, the UK, and Greece it is over 65% [3–7]. The worst situation is in Northern Ireland, where the population uses almost exclusively surface water resources for consumption purposes [8]. The highest concentrations of pesticides have been found in the runoff of meltwater or flood waters and agrochemical treatments [9]. A natural consequence of the presence of pesticides in surface water is their penetration into groundwater. Since surface and underground waters are a source of drinking water, they should not contain harmful substances. However, toxicological and epidemiological studies provide more and more evidence that the quality of some natural water poses a real threat to human

health. Among the several thousand organic compounds identified as water pollutants, chlorophenoxy herbicides make a significant contribution. This group of compounds is widely used to control broadleaf weeds in agricultural and non-agricultural areas, while 2,4-dichlorophenoxyacetic acid (2,4-D) and 4-chloro-2-methylphenoxyacetic acid (MCPA) are in the top five most often used herbicides in the US and the EU [10,11].

The chemical structures and the most relevant physicochemical properties of the chlorophenoxy herbicides used are presented in Table 1.

**Table 1.** The structure and physicochemical parameters of 2,4-D and MCPA.

| Parameter | 2,4-D | MCPA |
|---|---|---|
| CAS number | 94-75-7 | 94-74-6 |
| Chemical structure |  |  |
| Molecular weight (g mol$^{-1}$) | 221.04 | 200.62 |
| Molecular size (Å) | $4.88 \times 9.49$ | $5.47 \times 9.52$ |
| Solubility in water at 20 °C (g L$^{-1}$) | 0.682 | 0.825 |
| Melting point (°C) | 136–140 | 108–112 |
| Boiling point (°C) | 160 | 160 |
| pK$_a$ | 2.98 | 3.14 |
| log K$_{o/w}$ | 2.37 | 3.25 |

The permeation of chlorophenoxy herbicides into the body increases the risk of cancer, such as soft-tissue sarcoma, non-Hodgkin's lymphoma, and prostate and gastric cancers [12–15]. Other available data indicate correlations between exposure to these herbicides and Parkinson's disease and birth defects [16]. The toxicity of 2,4-D and MCPA is evidenced by the fact that the International Agency for Research on Cancer (IARC) has defined them as possibly carcinogenic to humans (Group 2B) [17]. For this reason, both the World Health Organization (WHO) and the United States Environmental Protection Agency (US EPA) have imposed requirements for water intended for human consumption, specifying that it should not contain more than 0.1 µg L$^{-1}$ of individual chlorophenoxy herbicides, and the total concentration should not exceed 0.5 µg L$^{-1}$ [18,19]. Such rigorous standards make the water purification process involving these types of compounds extremely important, from a scientific and practical point of view.

The techniques used for this purpose are diverse, and depending on the specificity of the purification process they are classified as physical, chemical, and biological methods, or combined ones. The most used water treatment techniques comprise: adsorption [20], membrane filtration [21], irradiation [22], photocatalytic degradation [23], thermal remediation [24], electrokinetic coagulation [25], oxidation [26], ozonation [27], chemical coagulation–flocculation [28], ion exchange [29], and biodegradation [30]. These techniques have their advantages and drawbacks. Sometimes the latter limits the usefulness of a given technique, especially in technological solutions used on a large scale. The most problematic disadvantages are: (i) high cost, (ii) high energy consumption, (iii) the need for chemical reagents, (iv) the creation of biosolids, sludge, or new harmful compounds as by-products, (v) the pre-treatment stage, (vi) membrane fouling, and (vii) the need for specific instruments and materials [22,26,30–32]. Among the above-mentioned techniques for water treatment, adsorption seems to be the most suitable because it is characterized by low cost and low energy consumption; moreover, it is highly selective and effective. There are various groups of adsorbents, i.e., (i) carbonaceous materials (commercial activated carbons, activated carbons from solid waste, carbon blacks, carbon nanotubes, graphene, reduced graphene oxide, and ordered mesoporous carbons); (ii) inorganic materials (silica gels, activated alumina, zeolites, molecular sieves, and metal oxides); (iii) advanced

materials (inorganic–organic composites, doped materials); and (iv) low-cost materials (agricultural and household waste-based materials, natural materials, biosorbents, industrial by-products) [33,34]. Due to the low cost of production for most of them, and the renewability and availability of the raw material, adsorbents based on agricultural and household waste are considered to be one of the most promising types. Taking into account these factors, there is no need for the subsequent regeneration and reusability of adsorbents based on agricultural and household waste. Additionally, the idea on the production of this type of adsorbents is also a solution for the proper management of agricultural and household waste. This is extremely important from an economic point of view, because the regeneration process requires energy or the use of expensive reagents. In general, agricultural and household waste can be used as adsorbents in the form of unmodified material, or after physical and/or chemical modification, as modified material, ash, biochar, or activated carbon. The latter, due to its relatively high efficiency in the purification of water from organic pollutants, in the near future, may completely replace commercial activated carbon, which comes from fossil coal and is characterized by a high market price. Straw, bark, and seed husks from various plants, seeds, and fruit peelings are just a few examples of renewable and readily available sources of raw material for activated carbon production. In terms of biochemistry, they mainly consist of biopolymers of cellulose and other polysaccharides (substances that are a source of elemental carbon), which makes them a suitable precursor of activated carbon. However, the production of this type of adsorbent is the most costly compared to other waste-based ones.

In the present paper, an overview of different adsorbents, based on agricultural and household waste, for chlorophenoxy herbicides removal from water is presented. The adsorbents were divided into several groups, depending on the applied method of modification used on the initial material and the type of final product obtained. The characteristics of the adsorbents, the conditions of the adsorption experiments, and the results, along with the theoretical models best describing the adsorption process, were reported. The analysis of the adsorption data, based on the adsorption capacity towards 2,4-D and/or MCPA and the adsorption rate, allowed us to indicate a group of the most effective adsorbents. The reviewed papers were also evaluated for the presence of studies that are crucial for the development of new adsorbents, such as: (i) the selectivity of the adsorbent; (ii) the adsorbent performance in multi-component systems (with heavy metal ions, antibiotics, phenols, dyes, competitive pesticides or metabolites of degradation, natural organic matter); (iii) herbicide interactions with adjuvants as integral components of herbicide formulations; (iv) the regeneration and reusability of the adsorbent; and (v) the cost balance of adsorbent production (collecting, transporting, and processing of waste) and the benefits from its use in the water treatment process.

Because many aspects have not been touched on at the preliminary research stage, the cost effectiveness of the production and applicability of adsorbents based on agricultural and household waste in the water treatment process is a debatable issue. Nevertheless, the authors hope the paper facilitates readers in the optimal selection of raw materials, the modification method particular to a given type of adsorbent, and the adsorption conditions. The paper also indicates the directions for future research that require development in the area of new materials dedicated to adsorption applications.

## 2. Adsorbents Based on Agricultural and Household Waste

The issue of water purification from chlorophenoxy herbicides is related to the selection of appropriate methods and materials for this purpose, which are effective, economical, and meet the requirements of "Green chemistry" (maximum reduction in the use and generation of harmful substances). The method that mostly meets these expectations is the adsorption of materials based on agricultural and household waste. Due to the biochemical structure, agricultural and household waste can be used as adsorbents in raw form, or after physical and/or chemical modifications, or they can be used as a precursor for the production of activated carbon. The effectiveness of chlorophenoxy herbicide adsorption

by these types of adsorbents depends on the pollutant concentration, the physicochemical characteristics of the adsorbent, and the conditions of the adsorption process (solution pH, temperature, ionic strength, and the accompanying substances in the system). In addition to the pollutant concentration, the degree of water purification is determined by the affinity of the herbicide to the adsorbent. Therefore, it is important to select the appropriate raw material and a method of modification/processing to obtain an effective adsorbent. Evaluation of the adsorption properties of the material is carried out based on the adsorption capacity and herbicide adsorption rate. Due to the complex composition of water from natural sources (the presence of other pollutants, i.e., competitive pesticides or metabolites of degradation, heavy metal ions, phenols, dyes, natural organic matter), it is also advisable to assess the selectivity of the adsorbent to a herbicide in multi-component systems and determine the effectiveness of chlorophenoxy herbicides to adsorption under such conditions. Commercial herbicide formulations include active substances and adjuvants that improve the solubility of the preparation, therefore determining the effect of auxiliary substances on the adsorption yield is equally important. With regard to activated carbon, the production of which is relatively expensive compared to other waste-based adsorbents, a study on material regeneration and reusability is recommended. In the final assessment on the potential use of adsorbents based on agricultural or household waste, it is also helpful to make use of the cost balance in adsorbent production (collecting, transporting, and processing of waste) and a list of benefits from the use of adsorbents for the removal of phenoxyacetic herbicides from water. Special attention has been paid to all these issues during the browsing of the cited papers.

### 2.1. Unmodified, Modified Natural Materials and Ashes

The development of adsorbents using agricultural and household waste provides an interesting alternative to commercial materials intended for environmental applications and at the same time provides a way for proper waste management. So far, these types of waste have been disposed of in landfills or by burning in open agricultural fields. Both methods release greenhouse gases that get into the atmosphere, while field burning is additionally associated with the destruction of the ecosystem and the risk of the uncontrolled spread of fire. When considering agricultural and household waste as raw materials for the preparation of adsorbents, one should emphasize its availability in huge quantities and its little or no economic value. All this means that more and more studies have been focused on the transformation of resource waste into useful and functional adsorbents for the removal of a variety of pollutants from water and wastewater. Excluding waste-based biochars and activated carbons, the remaining types of adsorbents have been obtained through simple physical or chemical modifications, several-stage and more complex modifications, as well as oxidative combustion. There have also been attempts to use unmodified waste materials. The usefulness of adsorbents in polluted water treatments was closely correlated with the method used for its preparation (or modification abandonment) and the type of raw material used.

One such typical agricultural waste is rice husk, which has been tested to remove phenoxy herbicides from water [35–37]. Deokar et al. described the liquid-phase removal of 2,4-D [35,38] and MCPA [37] using rice husk ash (RHA). The effects of various parameters, such as the adsorbent dosage, initial concentration, contact time, and particle size of the RHA, were studied. The adsorption kinetics were studied by applying the PFO, PSO, and Elovich equations, and the PSO and Elovich kinetic models were found to be the best models to describe the adsorption. The Langmuir, Freundlich, and Temkin isotherm models were considered to determine the adsorption capacity and adsorbent affinity. The maximum adsorption capacity, based on the Langmuir model, was found to be 1.425 mg g$^{-1}$ for 2,4-D and 1.681 mg g$^{-1}$ for MCPA. It was realized that the ash physicochemical parameters, such as a higher surface area and a smaller silica-to-carbon ratio, favored pollutant adsorption.

In other works [36,39], rice husk waste was reduced to nanosized particles via a mechanical method and was evaluated for its effective usage as a 2,4-D adsorbent. The

adsorption isotherms and kinetic models were used to describe the experimental data and understand the adsorption mechanism. From the results, the PSO model was found to fit the adsorption best; moreover, the adsorption of 2,4-D onto nanosized rice husk was the result of a combination of film diffusion and intraparticle diffusion processes. The Langmuir, Freundlich, Temkin, and DR isotherms were tested, and it was found that the Langmuir model fitted best with the adsorption data, with the maximum adsorption capacity of 24.75 mg g$^{-1}$ (in 2,4-D nanoformulation preparation experiments). In the case of the batch adsorption experiment, the Freundlich isotherm model was indicated as the most suitable one, with $K_F$ = 8.7 (mg g$^{-1}$)(L mg$^{-1}$)$^{1/n}$.

Okumuş et al. [40] investigated the potential of various bio-waste materials, including apple shell (AS), orange peel (OP), banana peel (BP), and millet waste (MW), as adsorbents for the removal of 2,4-D, 2,4-DP (2,4-dichlorophenoxy propanoic acid), and 2,4-DB (2,4-dichlorophenoxy butyric acid) from aqueous solutions. Batch adsorption experiments were conducted to study the effect of different operational parameters, such as the pH of the solution, the contact time, the adsorbent dosage, and the initial herbicide concentrations. The results showed that the optimum pH for the adsorbent was 6–7 and that the contact time was as short as 60 min for all of the adsorbates. The Langmuir and Freundlich models were applied to describe the adsorption isotherm, and the data fitted well with the Langmuir isotherms. The maximum adsorption capacity was obtained for 2,4-D as OP > MW > BP > AS; 2,4-DP as MW > AS > BP > OP; and 2,4-DB as OP > AS > BP > MW.

A study by Al-Zaben and Mekhamer [41] describes the use of coffee waste to remove MCPA from aqueous solutions. The effects of contact time, pH, and initial herbicide concentration were studied. The adsorption kinetics were well described by the PSO kinetic model, while the equilibrium results were better described by the Langmuir model. The equilibrium adsorption capacity was found to be 340 mg g$^{-1}$, using the Langmuir equation.

Trivedi et al. [42] published a research article that presented data about the use of cotton plant ash as a cheap adsorbent for the removal of 2,4-D. The BET surface area of the material was found to be 2 m$^2$ g$^{-1}$. The PFO, PSO, and Weber–Morris kinetic models were applied to the experimental data, and the PSO equation showed the best fit for the adsorption of 2,4-D. The equilibrium data were analyzed using Freundlich and Langmuir adsorption isotherms, and these fit well with the Langmuir model. Langmuir's adsorption capacity was found to be 0.64 mg g$^{-1}$.

The same authors [43] evaluated mustard plant ash as an adsorbent for the removal of 2,4-D from water. Optimum adsorption conditions were determined as a function of the adsorbent dose, the initial 2,4-D concentration, the contact time, and the temperature. Data from the experiments were fitted to various kinetic (PFO and PSO) and isothermal (Langmuir, Freundlich, and Temkin) models. The PSO kinetic model was found to show the best fit with the experimental data. The equilibrium data fitted best with the Langmuir adsorption isotherm, which confirmed the monolayer adsorption ($q_m$ = 0.76 mg g$^{-1}$). Thermodynamic constants indicated the spontaneity, as well as the exothermic nature, and irreversibility of the 2,4-D adsorption.

The next papers by this research team deal with the investigation of 2,4-D adsorption by groundnut shell ash [44,45] and wheat straw ash [46]. The first of these works [44] describes the herbicide adsorption of groundnut shell ash, with a BET surface area of 22 m$^2$ g$^{-1}$. Batch adsorption experiments were conducted to study the effect of different operational parameters, such as the adsorbent dosage, the contact time, and the initial adsorbate concentration, on the adsorption process. The PSO model, as well as the Langmuir adsorption isotherm model, showed the best fit with the experimental data. The maximum uptake capacity of the ash was determined as 0.87 mg g$^{-1}$. The paper [45] describes the preparation of three types of adsorbents produced using groundnut shells, namely ash, biochar, and activated carbon, and their adsorption performance.

Wheat straw ash was characterized and tested as a 2,4-D adsorbent [46]. The BET surface area of the material was found to be 37 m$^2$ g$^{-1}$. The effects of different operational parameters, such as the adsorbent dose, initial 2,4-D concentration, contact time, and pH,

were investigated. The experimental data were analyzed using the PFO and PSO kinetic models, as well as the Freundlich, Langmuir, and Temkin isotherm models. The PSO and Langmuir models showed the best fit with the experimental data. The adsorption capacity was found to be 1.89 mg g$^{-1}$.

Recently, the adsorption of 2,4-D and ketoprofen by *Physalis peruviana* fruit residue functionalized with sulfuric acid was reported by Dhaouadi et al. [47]. Adsorption isotherms were determined at 298–328 K and pH 2. It was found that the adsorption of 2,4-D by this biosorbent was multi-anchoring.

Peanut (*Arachis hypogaea*) skins treated with sulfuric acid were employed as an adsorbent for the removal of 2,4-D from aqueous solutions [48]. The results revealed that the adsorption was favored at acid pH = 2, and the optimal adsorbent dosage was 0.9 g L$^{-1}$. The equilibrium was reached quickly in the first 30 min, and the kinetics were well represented by the general order model. The Langmuir model was the one that obtained the best fit with the experimental data. The adsorption capacity of the acid-treated peanut skin towards 2,4-D was 246.72 mg g$^{-1}$.

In another work, wheat husks (*Fagopyrum esculentum*) modified using a treatment with $H_2SO_4$ were applied as an adsorbent to remove the 2,4-D herbicide from water [49]. The kinetics, equilibrium, and thermodynamic behavior, as well as the pH effects and adsorbent dosage, on the adsorption capacity were investigated. It was found that the Bangham kinetic model better represented the experimental data, and that the maximum adsorption capacity was 161.1 mg g$^{-1}$ at 298 K. The thermodynamic studies indicated spontaneous, favorable, and exothermic adsorption.

Recently, N-cetylpyridinium-modified tiger nut residue (TNR-CPC) as an adsorbent was investigated by Kani et al. [50]. The results revealed that the solution pH, adsorbent dose, temperature, and solution ionic strength affected the adsorptive capacity of TNR-CPC. The adsorption kinetics followed the PSO model, while the equilibrium data were best fitted to the Langmuir isotherm equation. The maximum monolayer adsorption capacity at 298 K was 79.3 mg g$^{-1}$.

Bartczak et al. [51] studied the use of saw-sedge Cladium mariscus (*C. mariscus*) for the adsorption of 2,4-D from aqueous systems. The *C. mariscus* consisted of carbon (48%), hydrogen (7%), nitrogen (0.95%), and sulfur (0.4%), and was characterized by a low surface area (0.6 m$^2$ g$^{-1}$). It was found that the adsorption performance was dependent on the following parameters: contact time, adsorbent mass, pH, and temperature of the adsorption system. Maximum adsorption was attained at pH = 3 and T = 25 °C, and equaled 65.58 mg g$^{-1}$. The experimental data corresponded to the pseudo-second-order kinetic model and the Langmuir isothermal model.

Recently, reports on the development and application of more advanced waste-based materials with magnetic properties for the removal of 2,4-D were published [52,53]. According to the authors of the paper [52], the forming process of a biomass-MOF composite was based on the loading of $NH_2$-MIL-101 (Fe) onto magnetized peanut husk. The adsorption capacity of the composite for 2,4-D removal was found to be 79.2 mg g$^{-1}$ according to the Langmuir isotherm model, while the kinetics were very rapid (attaining equilibrium within 5 min) and followed the PFO and PSO models. The second advanced magnetic material, a ferrospinel composite [53], was prepared based on activated carbon from lentil residues. For the synthesis of activated carbon, microwave-assisted $K_2CO_3$ chemical activation was applied instead of conventional heating, as a faster and more economical method. The analysis of the 2,4-D adsorption data indicated that the Langmuir model, with the maximum adsorption capacity of 400 mg g$^{-1}$ at 45 °C, and the pseudo-second-order model were the most suitable models. The magnetic properties of both the MOF and ferrospinel composites facilitated their separation from a purified medium.

Various unmodified and modified agricultural and household waste materials and ashes, and their adsorption performance in regard to chlorophenoxy herbicides removal, are summarized in Table 2. Additionally, the cost–benefit of the given adsorbents was evaluated based on a comparison between their adsorption capacity towards chlorophe-

noxy herbicides and the synthesis cost, the impact of their production on the natural environment (energy consumption, harmfulness of the reagents used, secondary waste), and their recyclability. Such subjective evaluation is widely accepted and practiced by many scientists [54,55].

To sum up, due to the variety of materials (i.e., pure agricultural and household waste, physically or chemically modified materials, ashes, and advanced composite materials), they show diversity in their biochemical composition, texture, chemical structure, and surface chemistry. The majority of adsorbents have a non-porous structure and their ability to bind herbicides from aqueous solutions is closely correlated with the presence of various functional groups of natural origin, introduced on their surface during the modification process, or resulting from the oxidation of mineral compounds during raw material combustion. Ashes are characterized by the lowest adsorption capacity, ranging from 0.64 to 1.89 mg g$^{-1}$ and, therefore, their adsorption potential is considered secondary to other functions they can perform in the economy and agriculture. First of all, the generation of ash involves the use of waste with low moisture content as feedstock for industrial boilers or home heating furnaces. Thus, the product obtained is not an end in itself, but is a consequence of the functioning of the fuel economy. Due to its rich composition of micronutrients, such as Ca, Mg, K, P, and Si, its high water capacity, and its carbon sequestration capability, ash is considered a useful addition to soil, improving its fertility. Moreover, ash after being spread on fields plays a protective role for the soil, as it adsorbs herbicides. The adsorption is relatively low, but equilibrium is reached almost instantly.

The adsorption capacities of unmodified, physically modified, and chemically modified materials are in the range of 22.71–340 mg g$^{-1}$, 24.75–76.92 mg g$^{-1}$, and 79.3–246.7 mg g$^{-1}$, respectively. Taking into account the simplicity and low cost of production of these adsorbents, or even the possibility of using some waste in an unprocessed form, these results seem to be promising.

Regarding magnetic composite materials, their adsorption capacities are equal to 79.2–400 mg g$^{-1}$. The great advantage of these adsorbents is the ease of their separation, from the medium subjected to purification, using an external magnet.

**Table 2.** The results from chlorophenoxy herbicides adsorption studies on unmodified adsorbents, modified adsorbents, and ashes based on agricultural and household waste.

| Adsorbent | $S_{BET}$ m$^2$ g$^{-1}$ | Adsorption Capacity ($q_m$) mg g$^{-1}$ | | Isotherm Model | Kinetic Model | Integral Cost–Benefit | Ref. |
| | | 2,4-D | MCPA | | | | |
|---|---|---|---|---|---|---|---|
| Rice husk ash | 34 | 1.425 | - | L, F, Te | PFO, PSO, E | Low | [35] |
| Rice husk ash | 34 | - | 1.681 | L, F, Te | PFO, PSO, E | Low | [37] |
| Nanosized rice husk | - | 24.75 | - | L, F, Te, DR | PFO, PSO, W-M | Medium | [36] |
| Nanosized rice husk | - | 76.92 | - | L, F, Te, DR | PFO, PSO, W-M | Medium | [39] |
| Cotton plant ash | 2 | 0.64 | - | L, F | PFO, PSO, W-M | Low | [42] |
| Mustard plant ash | - | 0.76 | - | L, F, Te | PFO, PSO | Low | [43] |
| Groundnut shell ash | 22 | 0.87 | - | L, F, Te | PFO, PSO | Low | [44] |
| Groundnut shell ash | 8 | 0.87 | - | L, F, Te | PFO, PSO | Low | [45] |
| Wheat straw ash | 37 | 1.89 | - | L, F, Te | PFO, PSO | Low | [46] |
| Functionalized *Physalis peruviana* biomass | - | 233.3 | - | ITM | - | High | [47] |

**Table 2.** *Cont.*

| Adsorbent | $S_{BET}$ m$^2$ g$^{-1}$ | Adsorption Capacity ($q_m$) mg g$^{-1}$ | | Isotherm Model | Kinetic Model | Integral Cost–Benefit | Ref. |
|---|---|---|---|---|---|---|---|
| | | 2,4-D | MCPA | | | | |
| Acid-treated peanut skin | - | 246.7 | - | L, F, Te, DR | PFO, PSO, E | High | [48] |
| Apple shell | - | 40.08 | - | L, F | Contact time | Medium | [40] |
| Orange peel | - | 22.71 | - | L, F | Contact time | Medium | [40] |
| Banana peel | - | 33.26 | - | L, F | Contact time | Medium | [40] |
| Millet waste | - | 45.45 | - | L, F | Contact time | Medium | [40] |
| Coffee wastes | - | - | 340.0 | L, F | PFO, PSO | High | [41] |
| N-cetylpyridinium-modified tiger nut residue | 0.03 | 79.3 | - | L, F, K-C | PFO, PSO, E, W-M | Medium | [50] |
| H$_2$SO$_4$-modified wheat husks | - | 161.1 | - | L, F | Ba | High | [49] |
| Saw-sedge Cladium mariscus | 0.6 | 65.58 | - | L, F | PFO, PSO | Medium | [51] |
| Biomass-based MOF composite | 71 | 79.2 | - | L, F, Te, K-C | PFO, PSO | Medium | [52] |
| Biomass-based ferrospinel composite | 528 | 400 | - | L, F | PFO, PSO, W-M | High | [53] |

### 2.2. Biochars

Biochar is a solid product that can be obtained from the thermal degradation of agricultural and household waste under oxygen-limited conditions (pyrolysis process). Biochar is generated from the part of biomass that has already been used in human activities and is problematic for society due to the need for storage and the putrefaction that occurs within it, which may lead to the uncontrolled emission of greenhouse gases. The physicochemical properties of biochar greatly depend on various parameters, such as the biomass type, pyrolysis temperature and duration, pyrolysis procedure, and post-treatment processes. It is generally assumed that biochar is produced at low temperatures, but in practice a wide temperature range of 200–800 °C has been used. In the course of biochar production, the volatile constituents and tar existing in the biomass are emitted and, as a result, a porous structure in the final product is formed. The development of the porous structure is visible in the specific increase in the surface area of the biochar by up to several folds compared to that of ashes. The physicochemical properties of the final product determine its suitability for specific applications. From the perspective of the utilization of biochar as an adsorbent, a high capacity towards pollutants, fast adsorption kinetics, and low reversibility of the adsorption process are the most desirable features. In recent years, several studies in the scientific literature have dealt with the production of designed biochars suitable for the adsorption of various organic pollutants, including phenoxyacetic herbicides.

Mandal et al. [56] studied the adsorption process of 2,4-D by biochars produced from various green wastes, namely tea (TW), oak wood (OW), bamboo (B), and bur cucumber (BU), at two pyrolytic temperatures (400 and 700 °C). The specific surface area of the TW, OW, BU, and B biochars was 421.3, 270.7, 475.6, and 2.3 m$^2$ g$^{-1}$, respectively. The experimental data were fitted using the Langmuir and Freundlich isotherm models, as well as the PFO, PSO, and Elovich kinetic models. The kinetics of the 2,4-D adsorption was best described by the PSO kinetic model, and the adsorption equilibrium data were successfully fitted to both the Langmuir and Freundlich equations. The maximum adsorption capacity was 10.05, 42.67, 26.66, and 28.92 mg g$^{-1}$ for TW, BU, OW, and B biochars, respectively.

A carbonized chestnut shell was used for the removal of 2,4-D from water [57]. The surface area of the carbonized adsorbent was 280.4 m$^2$ g$^{-1}$. The adsorption equilibrium was achieved in 250 min, and the kinetics followed the PFO model. The experimental data were modeled using the Langmuir, Freundlich, Temkin, and Dubinin–Radushkevich equations. The Temkin isotherm model represented the adsorption phenomenon very well, at all the temperatures studied (at 35, 45, and 55 °C). Langmuir's maximum adsorption of 2,4-D was 0.93 mg g$^{-1}$.

Biochars generated from corncobs, bamboo, wood chips, and rice straw were tested as adsorbents for the removal of 2,4-D from aqueous solutions [58]. Various pyrolysis conditions, including the pyrolysis time and temperature (400–700 °C), were investigated. The highest BET surface area was 389 m$^2$ g$^{-1}$ (700 °C) for the wood chips biochar, 510 m$^2$ g$^{-1}$ (700 °C) for the bamboo biochar, 128 m$^2$ g$^{-1}$ (500 °C) for the rice straw biochar, and 2.3 m$^2$ g$^{-1}$ (600 °C) for the corncobs biochar. The Freundlich K$_F$ adsorption parameters were 8.72, 19.5, 1.23, and 1.06 (mg g$^{-1}$)(L mg$^{-1}$)$^{1/n}$, respectively.

The corncob biochar was synthesized at 600 °C for 4 h, as a low-cost adsorbent for the removal of 2,4-D from water [59]. The specific surface area of the biochar was 298 m$^2$ g$^{-1}$. It was found that the pH was a key factor for 2,4-D removal by the corncob biochar. The experimental adsorption data followed the PSO model and the Langmuir isotherm, with a maximum adsorption capacity of 37.40 mg g$^{-1}$.

A study by Essandoh et al. [60] describes the preparation of biochar from switchgrass (*Panicum virgatum*). The BET surface area of the biochar was found to be 1.1 m$^2$ g$^{-1}$. Kinetic and adsorption isotherm studies for 2,4-D and MCPA were conducted. The PSO model was superior to the PFO kinetic model for both 2,4-D and MCPA. The adsorption isotherms were described by the Freundlich, Langmuir, Toth, and Redlich–Peterson models. In general, all the theoretical isotherm equations provided reasonable fits with the experimental data; however, the highest overall regression coefficients belonged to the data fitted to the Freundlich model (in the case of 2,4-D) and the Redlich–Peterson model (in the case of MCPA). The maximum adsorption capacity for 2,4-D and MCPA was 134 mg g$^{-1}$ and 50 mg g$^{-1}$, respectively.

A series of biochars were produced from softwood shavings (WS), pig manure (PM), and sewage sludge (SS), under various conditions (with an initial heating rate of 25 °C min$^{-1}$ up to 200 °C, 350 °C, and 500 °C, maintained for 4 h) [61]. The specific surface area ranged from 4 to 347 m$^2$ g$^{-1}$ for the WS, from 2 to 27 m$^2$ g$^{-1}$ for the PM, and 11 to 43 m$^2$ g$^{-1}$ for the SS. The biochars were examined as potential adsorbents for the removal of 2,4-D, MCPA, 2,4-DB (4-(2,4-dichlorophenoxy)-butanoic acid), and triclosan (5-chloro-2-(2,4-dichlorophenoxy)phenol) from an aqueous solution. The Freundlich model fitted the adsorption isotherm of the biochars very well. The results revealed that the adsorption increased in the order: 2,4-D < MCPA < 2,4-DB < triclosan.

Biochars produced by slow pyrolysis from a mixture of about 80% birchwood (*Betula* sp.) and 20% Norway spruce wood (*Picea abies*) were tested in batch [62] and column [63] studies. The wood-based biochar was additionally modified by a heat treatment and by an iron salts treatment [62]. The BET surface area of the untreated biochar, heated biochar, and biochar treated with iron salts was 74, 116, and 59 m$^2$ g$^{-1}$, respectively. These materials were used for the adsorptive removal of five pesticides, including bentazone, chlorpyrifos, diuron, glyphosate, and MCPA, from aqueous solutions. The experimental data were modelled using the Freundlich isotherm model, and the K$_F$ values decreased in the order: diuron > chlorpyrifos > MCPA > bentazone > glyphosate. The results showed that heat treatment increased the adsorption of bentazone and MCPA, while iron salts treatment increased the adsorption of glyphosate.

The Liu research group [64] studied the adsorption performance of hydrochars (HCs) prepared by the hydrothermal carbonization (HTC) of lettuce waste (LHC), taro waste (THC), and watermelon peel (WHC) in low-temperature conditions (180–240 °C). The determined adsorption capacities of the HCs were hydrochar precursor and HTC temperature dependent. The greatest capacities of LHC$_{180}$ (88.4 mg g$^{-1}$), THC$_{180}$ (90.2 mg g$^{-1}$), and WHC$_{240}$

(88.4 mg g$^{-1}$) for 2,4-D removal were achieved at HTC temperatures of 180, 180, and 240 °C, respectively. It was found that such hydrochar properties as a well-developed surface area with small mesopores, low aromaticity, and a high amount of C–O functional groups, favored herbicide adsorption via intensive partitioning and/or chemisorption. The pseudo-second-order model fitted the 2,4-D adsorption data better than the pseudo-first-order model.

A series of biochars based on water lettuce (WL), water hyacinth (WH), wheat bran (WB), pinewood sawdust (PS), and oak tree wood biomass (OTW), was fabricated via pyrolysis at 500 °C [65]. The adsorption performance was found to be in the following order: PS (62.53%) > OTW (53.24%) > WB (46.56%) > WH (33.54%) > WL (28.25%).

Ma et al. [66] and Almahri et al. [67] studied spent coffee ground biochars as adsorbents for the removal of 2,4-D from aqueous solutions. Ma et al. [66] obtained the char via low-temperature pyrolysis (300 °C) and impregnation with $H_3PO_4$ as a pre-treatment stage. The best kinetic and isotherm fits were found for the Elovich and Freundlich equations, respectively. It was shown that the adsorption efficiency diminishes with a rising pH, and the highest efficiency was achieved at a pH of 2. The maximum adsorption capacity of biochar was reported as 323.76 mg g$^{-1}$. The thermodynamic studies pointed out that the adsorption process was exothermic. Almahri et al. [67] synthetized biochar by biomass pyrolysis at 700 °C in a neutral atmosphere, followed by the transfer to a steam atmosphere. The surface area and pore volume of the biochar reached 422.4 m$^2$ g$^{-1}$ and 0.46 cm$^2$ g$^{-1}$, respectively. The results showed that the adsorption behavior was significantly impacted by changes in the solution pH. The pH value of 6 was indicated as the most beneficial to the adsorption process ($q_m$ = 276.3 mg g$^{-1}$). The thermodynamic parameters were determined and revealed that the process was endothermic, chemisorptive, and spontaneous. The adsorption process was best fitted to the Langmuir isotherm model and the pseudo-second-order kinetic model.

In the subsequent works [68,69], the adsorption of 2,4-D by rice husk biochar from synthetic wastewater and drainage water was studied. Of the various operating parameters, the initial adsorbate concentration and adsorbent dosage were indicated as those that had the greatest impact on the adsorption capacity. The highest efficiency of biochar in 2,4-D removal was achieved at a pH of 5.5, a contact time of 80 min, a temperature of 60 °C, an adsorbent dose of 0.2 g, and an initial solution concentration of 60 mg L$^{-1}$. The PFO and PSO kinetic models, as well as the Langmuir, Freundlich, Langmuir–Freundlich, and Redlich–Peterson isotherms, were tested. The pseudo-first-order model and Freundlich equations were chosen as those that best described the adsorption process. The maximum adsorption capacity of 2,4-D by rice husk biochar equaled 246 mg g$^{-1}$. Regarding thermodynamics, the adsorption process was exothermic and spontaneous. The adsorption potential of the biochar was also evaluated with other materials, e.g., granular activated carbon (GAC) and multi-walled carbon nanotubes (MWCNTs). The maximum adsorption capacities of the chars were lower than the GAC, but higher than the MWCNTs. However, in terms of the herbicide removal dynamics, the adsorption process required the longest time to reach an equilibrium state.

A study by Zhu et al. [70] describes the effect of various modifications of corn stalk biochar (BC) on the adsorption properties of the obtained materials. The applied modifications involved activation of the BC with potassium carbonate (KBC), followed by surface oxidation (OKBC), surface amination (NKBC), loading nano-zero valent iron (nZVI@KBC), and loading nano-iron oxyhydroxide (nHIO@KBC), respectively. The biochars exhibited the following order towards 2,4-D adsorption capacity: NKBC > KBC > OKBC > BC > nHIO@KBC > nZVI@KBC. It was shown that a well-developed specific surface area of the adsorbent was not the only factor determining the adsorption capacity. The surface properties of the adsorbent, such as the polarity, hydrophobicity, and surface functional groups, are equally important factors. The activation of the biochar structure with potassium carbonate, surface amination, or surface oxidation, made it more suitable for 2,4-D adsorption in comparison with unmodified char, while the surface loading of nanometal and nanometal oxyhydroxide made the biochar worse regarding its adsorbency.

Kearns et al. [71,72] tested the usefulness of biochars generated from updraft gasifiers, under conditions of simultaneous co-pyrolysis thermal air activation (CPTA), on the adsorption of 2,4-D and simazine from surface water containing dissolved organic matter. Cherry pits, Jatropha press-cake waste, sugarcane bagasse pellets, pecan shells, bamboo, pine, eucalyptus, and longan wood, were used as raw materials. Enhanced adsorption capabilities of the as-received chars by about one order of magnitude, compared to conventional anoxic pyrolysis (CAP) biochars, were observed. Applying a high-temperature synthesis mode combined with CPTA conditions resulted in biochars with a specific surface area and mesopore surface area of 330 and 110 $m^2 g^{-1}$, respectively. Such characteristics of the final product were related to the thermochemical widening of the pores and/or the removal of the pyrolysis tars from its structure. Increasing the pyrolysis temperature or process duration during the biochar synthesis procedure, conducted by using a horizontal drum kiln, increased the 2,4-D adsorption capacity of the biochar, whereas the type of feedstock did not affect it.

The removal of 2,4-D and carbofuran from water by biochars obtained from agricultural by-products, i.e., rice husk (RH), corn stover (CS), corncob (CC), and sorghum stems (SS), was studied as a function of pH [73]. It was shown that the greatest efficiency in herbicide removal was obtained at pH = 6. The equilibrium adsorption capacities of these biochars for 2,4-D were found to be 0.64, 0.76, 0.81, and 0.71 $mg g^{-1}$ for RH, CS, CC, and SS, respectively.

Evaluation of the impact of the physicochemical properties of a high-temperature wheat straw biochar on its adsorption behavior related to various pesticides (2,4-D, MCPA, metolachlor, carbaryl, and carbofuran) was the research theme of Ćwieląg-Piasecka et al. [74]. The obtained adsorbent was characterized by a surface area of 237.39 $m^2 g^{-1}$. The adsorption affinity of the biochar to the pesticides decreased in the order: carbofuran > carbaryl > metolachlor > 2,4-D > MCPA. The relatively low adsorption capability of the biochar for chlorophenoxy acids was ascribed to the electrostatic attraction between the herbicides and the polar functional groups on the BC surface, or to a nonspecific interaction with the adsorbate. Hydrophobic interaction was postulated as the main mechanism of pesticide adsorption by the BC.

Lü et al. [75] tested the use of rice straw biochars made by slow pyrolysis at 200 °C (RS200), 350 °C (RS350), and 500 °C (RS500), for the regulation on the release and leaching of acetochlor and 2,4-D in soils. Part of the research concerned the equilibrium adsorption of herbicides by biochars differentiated in structure ($S_{BET}$ = 2.1, 20.6, and 128 $m^2 g^{-1}$, respectively), the elemental composition, and ash residue. The Freundlich $K_F$ adsorption parameters for the adsorption of 2,4-D were 0.206, 4.48 and 3.24 $(mg^{-1})(L mg^{-1})^{1/n}$ by RS200, RS350, and RS500 biochars, respectively. The lower adsorption capacity of RS500 than RS350, despite its more developed surface area, was explained by the high content of stable minerals (ash), accounting for nearly one-half of the mass of the sample.

Trivedi et al. [42,45,46] used the cotton plant biochar ($S_{BET}$ = 109 $m^2 g^{-1}$), groundnut shells biochar ($S_{BET}$ = 43 $m^2 g^{-1}$), and wheat straw biochar ($S_{BET}$ = 96 $m^2 g^{-1}$), for the removal of 2,4-D. The experimental adsorption data for the studied biochars followed the PSO model and the Langmuir isotherm, with a $q_m$ that equaled 3.93, 3.02, and 2.02 $mg g^{-1}$.

Table 3 shows some of the adsorption capacities of the biochars reported in the literature.

Most of the pristine biochars synthetized through pyrolysis under a neutral atmosphere are characterized by a relatively small surface area (~1–400 $m^2 g^{-1}$), an undeveloped pores structure, and a limited amount of surface functional groups. Only a few biochars have a more extensive network of internal pores, translating into surface areas in the range of 400–523 $m^2 g^{-1}$. However, the more developed pores' structure in a solid does not always guarantee its ability to adsorb chlorophenoxy acids from aqueous solutions. There are also biochars with a poor pores' structure that exhibit relatively high adsorption capacities for these pollutants. Noteworthy are the hydrochars whose adsorption capacity towards 2,4-D results mainly from the surface chemistry of the solid and to a lesser extent from its porosity. It has been found that the greater the aromaticity, the oxygen-containing

surface functional groups (e.g., phenolic, carboxyl, hydroxyls), and the lower the content of ash in the biochar, improves its adsorption ability. These solid features, in turn, depend on various factors, such as the type of raw material, the synthesis parameters, the furnace characteristics, and the post-synthesis treatments.

To enhance the applicability of biochars in removing chlorophenoxy herbicides from aqueous solutions, they were subjected to structural activation or surface modification. Structural activation was carried out using: (i) specialized equipment, e.g., an updraft gasifier working under conditions of simultaneous co-pyrolysis and thermal air activation, (ii) gases, or (iii) chemicals. Regarding surface modification of biochars, the processes of amination and oxidation brought satisfactory results. In our opinion, the structural activation of biochar using a steam atmosphere or potassium carbonate leads to a change in the classification of the final product from biochar to activated carbon, but they have been left in this subsection based on the nomenclature given by the researchers.

Some biochars exhibit adsorption capacities comparable to waste-based activated carbons (see the subsection below), but their production is low cost and does not use harmful chemicals, so it can be concluded that these materials are competitors to activated carbons in regard to the adsorption techniques for removing phenoxyacetic herbicides from water and wastewater.

In the case of those biochars that show relatively low adsorption capacity towards herbicides, they may be suitable materials for the prevention of herbicide leaking and mobility in soil profiles if they are applied as a soil amendment. Moreover, biochars have great potential for the improvement of soil properties, such as increasing the pH, nutrient availability, and water retention. These filter materials could be applied in farm fields, as well as in agricultural ditches, drainage systems, and places accidentally contaminated with chlorophenoxy herbicides. Mitigating climate change is the secondary environmental benefit of biochar soil amendment associated with carbon sequestration.

**Table 3.** The results from chlorophenoxy herbicides adsorption studies on biochars.

| Adsorbent | $S_{BET}$ m$^2$ g$^{-1}$ | Adsorption Capacity ($q_m$) mg g$^{-1}$ | | Isotherm Model | Kinetic Model | Integral Cost–Benefit | Ref. |
|---|---|---|---|---|---|---|---|
| | | **2,4-D** | **MCPA** | | | | |
| Cotton plant biochar | 109 | 3.93 | - | L, F | PFO, PSO, W-M | Low | [42] |
| Groundnut shell biochar | 43 | 3.02 | - | L, F, Te | PFO, PSO | Low | [45] |
| Wheat straw biochar | 96 | 2.02 | - | L, F, Te | PFO, PSO | Low | [46] |
| Tea waste biochar | 421 | 10.05 | - | L, F | PFO, PSO, E | Low | [56] |
| Oak wood biochar | 271 | 26.66 | - | L, F | PFO, PSO, E | Medium | [56] |
| Bamboo biochar | 476 | 28.92 | - | L, F | PFO, PSO, E | Medium | [56] |
| Bur cucumber biochar | 2.3 | 42.67 | - | L, F | PFO, PSO, E | Medium | [56] |
| Carbonized chestnut shell | 280 | 0.93 | - | L, F, Te, DR | PFO, PSO, W-M | Low | [57] |
| Corncob biochar | 298 | 37.40 | - | L, F | PFO, PSO | Medium | [59] |
| Switchgrass biochar | 1.10 | 134 | 50 | L, F, To, DR | PFO, PSO | High | [60] |
| Lettuce waste hydrochar (LHC$_{180}$) | 6.28 | 88.4 | - | - | PFO, PSO | Medium | [64] |
| Lettuce waste hydrochar (LHC$_{200}$) | 6.90 | 80.0 | - | - | - | Medium | [64] |
| Lettuce waste hydrochar (LHC$_{220}$) | 6.47 | 80.9 | - | - | - | Medium | [64] |

**Table 3.** *Cont.*

| Adsorbent | $S_{BET}$ m$^2$ g$^{-1}$ | Adsorption Capacity ($q_m$) mg g$^{-1}$ | | Isotherm Model | Kinetic Model | Integral Cost–Benefit | Ref. |
|---|---|---|---|---|---|---|---|
| | | 2,4-D | MCPA | | | | |
| Lettuce waste hydrochar (LHC$_{240}$) | 3.67 | 77.4 | - | - | - | Medium | [64] |
| Taro waste hydrochar (THC$_{180}$) | 9.23 | 90.2 | - | - | - | Medium | [64] |
| Taro waste hydrochar (THC$_{200}$) | 2.51 | 35.5 | - | - | - | Medium | [64] |
| Taro waste hydrochar (THC$_{220}$) | 3.12 | 47.5 | - | - | - | Medium | [64] |
| Taro waste hydrochar (THC$_{240}$) | 0.68 | 33.6 | - | - | - | Medium | [64] |
| Watermelon peel hydrochar (WHC$_{180}$) | 3.29 | 45.7 | - | - | - | Medium | [64] |
| Watermelon peel hydrochar (WHC$_{200}$) | 5.82 | 59.7 | - | - | - | Medium | [64] |
| Watermelon peel hydrochar (WHC$_{220}$) | 8.45 | 64.0 | - | - | - | Medium | [64] |
| Watermelon peel hydrochar (WHC$_{240}$) | 5.99 | 88.4 | - | - | - | Medium | [64] |
| Coffee ground biochar | 82 | 323.76 | - | L, F | PFO, PSO, E | High | [66] |
| Water lettuce biochar | - | 84.79 | - | L | - | Medium | [65] |
| Water hyacinth biochar | - | 100.62 | - | L | - | High | [65] |
| Wheat bran biochar | - | 139.69 | - | L | - | High | [65] |
| Pinewood sawdust biochar | - | 187.60 | - | L | - | High | [65] |
| Oak tree wood biochar | - | 159.69 | - | L | - | High | [65] |
| Coffee ground biochar | 422.4 | 276.3 | - | L, F, Te, DR, J, K, Hi, S | PFO, PSO, E, IDM, A | High | [67] |
| Rice husk biochar | 320 | 246 | - | L, F, L-F, R-P | PFO, PSO | High | [68,69] |
| Corn stalk biochar (BC) | 523 | 8.54 | - | L, F | - | Low | [70] |
| Modified corn stalk biochar (KBC) | 680 | 19.8 | - | L, F | - | Low | [70] |
| Modified corn stalk biochar (OKBC) | 290 | 12.87 | - | L, F | - | Low | [70] |
| Modified corn stalk biochar (NKBC) | 691 | 22.84 | - | L, F | - | Medium | [70] |
| Modified corn stalk biochar (nZVI@KBC) | 179 | 3.03 | - | L, F | - | Low | [70] |
| Modified corn stalk biochar (nHIO@KBC) | 272 | 6.53 | - | L, F | - | Low | [70] |
| Rice husk biochar | - | 0.64 | - | - | - | Low | [73] |
| Corn stover biochar | - | 0.76 | - | - | - | Low | [73] |
| Corncob biochar | - | 0.81 | - | - | - | Low | [73] |
| Sorghum stem biochar | - | 0.71 | - | - | - | Low | [73] |

### 2.3. Activated Carbons

Activated carbons are another group of adsorbents that can be obtained from agricultural and household waste and used for the removal of phenoxyacetic herbicides from water. Activated carbon synthesis involves two main stages: the carbonization of the raw material at temperatures below 800 °C in an inert atmosphere and the activation of the carbonized product (char). The latter stage is carried out by physical and/or chemical activation. Physical activation is based on the char treatment, with oxidizing gases at elevated temperatures. During this process, the char is partially gasified leading to the development of the internal surface of the solid. Chemical activation is based on the mixing of the raw material with a chemical reagent, i.e., oxidizing acid, salt solution, or alkali followed by heat treatment. The mechanism of chemical activation depends on the activating agent used and, in some cases, is very complex. Depending on the type of raw material, the activating agent, and the conditions of carbonization and activation processes, activated carbons may be characterized by various pore structures, surface chemistry, and adsorption properties towards specific pollutants.

The agricultural and household waste-based activated carbons presented in this subsection, as potential adsorbents for the water treatment of phenoxyacetic herbicides, were divided into groups depending on the type of activation process used. Among the physical activating agents, steam (ST) and carbon dioxide (CD) are the most popular; however, mixtures of these agents and microwave radiation (MV) are occasionally used. In one case, the activation process followed the self-activation mode, where various gases like $CO$, $CH_4$, $H_2$, $CO_2$, and $H_2O$, emitted during the biomass pyrolysis, were activating agents. Regarding chemical activation, the leading activating agents were zinc chloride ($ZnCl_2$), followed by potassium hydroxide (KOH), orthophosphoric acid ($H_3PO_4$), sulfuric acid ($H_2SO_4$), a combination of different chemical activating agents ($H_2SO_4$ and KOH; KOH and $Fe_3O_4$), or a combination of chemical and physical activating agents (KOH and $CO_2$).

### 2.3.1. Activated Carbons Based on Physical Activation

Activated carbon produced from *Lagenaria vulgaris* shell by thermo-chemical carbonization and steam activation was applied to the removal of 2,4-D from aqueous solutions by Bojić et al. [76]. The BET of the as-prepared AC was found to be 665 $m^2$ $g^{-1}$. The equilibrium data were fitted to the Langmuir, Freundlich, Sips, and Brouers–Sotolongo models, and the maximum adsorption capacity was 333.3 mg $g^{-1}$. The thermodynamic data showed that the adsorption was endothermic, spontaneous, and physical.

Brito et al. [77] evaluated the removal efficiency of 2,4-D from water by three ACs from agricultural biomass waste. These materials were prepared using the one-step carbonization/physical activation route, using steam at 800 °C under an argon atmosphere, from sugarcane bagasse, coconut shell, and endocarp from babassu coconut. The BET surface areas of the ACs were 547, 991, and 1068 $m^2$ $g^{-1}$, respectively. The adsorption process for 2,4-D was investigated using the Langmuir, Freundlich, Redlich–Peterson, Temkin, and Dubinin–Radushkevich isotherm models. The results revealed that the Langmuir and R-P equations best described the adsorption process. The adsorption kinetics were described well by the PSO model, and the thermodynamic data indicated the spontaneous nature of the adsorption process.

Amiri et al. [78] investigated the potential of activated carbon derived from canola stalk as an adsorbent for 2,4-D removal from water. The AC was synthesized via the physical activation of the canola stalk biochar by steam, and its BET surface area was 556.8 $m^2$ $g^{-1}$. According to the authors, the maximum amount of 2,4-D removal was achieved as 135.8 mg $g^{-1}$ under a pH of 2 and an initial herbicide concentration of 150 mg $g^{-1}$. The particle diffusion model was the best-fitting kinetic model for the 2,4-D adsorption.

Mandal et al. [56] studied the adsorption process for 2,4-D using activated carbon produced from tea biochar, through activation with steam. The specific surface area of the material was 576 $m^2$ $g^{-1}$. The herbicide adsorption followed the PSO kinetic model, and

the adsorption equilibrium data were successfully fitted to the Langmuir equation, with a maximum adsorption capacity of 58.8 mg g$^{-1}$.

Orduz et al. [79] investigated the 2,4-D adsorption by activated carbon from peanut shells (steam activation, $S_{BET}$ = 1240 m$^2$ g$^{-1}$) in aqueous and organic solvents. The pollutant desorption kinetics and the application of the AC as a filler in a solid-phase extraction procedure (SPE) were presented. It was stated that the pollutant affinity towards the AC was higher in aqueous solvents than in organic ones (acetonitrile and methanol), with q = 549 mg g$^{-1}$ (F). The 2,4-D desorption process was the most effective in methanol (regarding the pollutant amount and desorption rate); mini columns filled with AC were characterized by high reuse capacity.

Cansado et al. [80] obtained activated carbons based on *Tectona grandis* tree sawdust using physical activation with carbon dioxide at 973 K and different activation times, e.g., 360 and 480 min for Teak-7360 and Teak-7480, respectively. The adsorbents were characterized by high values for the specific surface area (787 and 910 m$^2$ g$^{-1}$) and were applied for the removal of MCPA during the liquid phase. The adsorption capacity of Teak-7360 and Teak-7480 was found to be 212.7 and 313 mg g$^{-1}$, respectively. The herbicide adsorption experimental data were optimized using the Langmuir and Freundlich equations.

The same gaseous activating agent in the production of activated carbons, based on wood waste from Angola, was used by Tchikuala et al. [81]. The activation time and carbon precursor were variable synthesis factors. It was stated that the values of MCPA adsorption (85–295 mg g$^{-1}$) correlated with the degree of development of the activated carbons' specific surface area (603–838 m$^2$ g$^{-1}$).

Blachnio et al. [82] used pistachio shells for the preparation of activated carbons through physical activation using carbon dioxide; steam and carbon dioxide; and carbon dioxide and microwave radiation. The obtained activated carbons were characterized by high values for the specific surface area (556–685 m$^2$ g$^{-1}$) and were utilized for the removal of the model organic pollutants from the single- and multi-component systems. The adsorption capacities for MCPA removal were equal to 314–288.9 mg g$^{-1}$ (the general Langmuir isotherm model). A slight effect of the crystal violet presence on the MCPA adsorption and the inverse of this was noticed as a result of the adsorption by different types of pores. For similar herbicides (MCPA and 2,4-D), strong competition in the capacity and adsorption rate was observed.

The paper [83] describes the results from the conversion of soybean residues via one-step microwave irradiation ($S_{BET}$ = 1696 m$^2$ g$^{-1}$) and its application for the removal of 2,4-D from aqueous solutions. The equilibrium uptake was analyzed using the Langmuir, Freundlich, and Temkin isotherms, with the Freundlich equation indicated as the best one (q = 253.17 mg g$^{-1}$). In addition to the adsorptive properties, the adsorbent showed antibacterial and antifungal activity.

Gurav et al. [65] fabricated a self-activated carbon from *Pinus taeda* wood by utilizing gases released during the biomass pyrolysis process. The as-received adsorbent with a well-developed surface area ($S_{BET}$ = 1450 m$^2$ g$^{-1}$) was applied for 2,4-D removal during the aqueous phase. At pH = 2, the maximum adsorption capacity reached 471.70 mg g$^{-1}$ (the Langmuir model). The adsorption kinetics obeyed the pseudo-second-order model. The effect of natural organic matter and salinity on 2,4-D removal was also tested. The reusability studies on self-activated carbon revealed outstanding performance in up to seven adsorption–regeneration cycles.

### 2.3.2. Activated Carbons Based on Chemical Activation

Activated carbon from sesame seeds was prepared by ZnCl$_2$ activation and its adsorption capacity for 2,4-D in aqueous solutions was investigated [84]. The effect of varying parameters (pH, temperature, adsorbent dose, and contact time) was also studied. The adsorption kinetics followed the PSO kinetic model. The Freundlich and DR models described the adsorption at equilibrium well.

In a study by Angin and Ilci [85], activated carbon was prepared from olive-waste cake by chemical activation using $ZnCl_2$ and used for 2,4-D removal. The results revealed that the adsorption kinetics and adsorption isotherms were described well by the PSO kinetic model and the Langmuir isotherm equation, respectively. The maximum adsorption capacity, based on the Langmuir model, was found to be 129.9 mg g$^{-1}$. The thermodynamic data indicated that the adsorption process was feasible, spontaneous, and exothermic.

The removal of 2,4-D from water using AC prepared from queen palm endocarp (*Syagrus romanzoffiana*) by pyrolysis with $ZnCl_2$ activation was also reported [86]. In that investigation, the influence of the initial pH, the adsorbent dose, the initial 2,4-D concentration, and the temperature on the adsorption was evaluated. The adsorption kinetic study indicated that the process reached equilibrium at 180 min and followed the PSO model. The equilibrium data were analyzed using the Langmuir, Freundlich, Temkin, and Dubinin–Radushkevich isotherm equations. The results showed that the adsorption isotherms were well described by the Langmuir equilibrium isotherm equation, and the calculated monolayer adsorption capacity was 367.8 mg g$^{-1}$. The thermodynamic data indicated that the adsorption process was spontaneous and endothermic.

A recent study by Angın and Güneş [87] describes the preparation of activated carbon from orange (*Citrus sinensis* L.) pulp by chemical activation with zinc chloride. The AC was characterized and then used as an adsorbent for 2,4-D removal from an aqueous solution. The effects of the adsorbent dosage, the contact time, the initial concentration of 2,4-D, the temperature, and the pH, on the adsorption were studied. The PSO, PFO, and intra-particle diffusion kinetic models were tested with the experimental data, and the PSO kinetics model was the best fit for the adsorption of 2,4-D by the orange pulp-derived AC. The experimental data were analyzed by the Langmuir, Freundlich, Temkin, and Dubinin–Radushkevich isotherm models. The equilibrium data fitted well with the Langmuir model, with a maximum adsorption capacity of 71.9 mg g$^{-1}$. The thermodynamic parameters indicated feasible, spontaneous, and exothermic adsorption.

Boumaraf et al. [88] reported the synthesis, characterization, and applicability of activated carbon from *Elaeagnus angustifolia* seeds using $ZnCl_2$ activation. The influence of the operating parameters (solid dose, contact time, stirring speed, and pH) on 2,4-D adsorption was investigated. The adsorption process was spontaneous and exothermic, and obeyed the Langmuir and the pseudo-second-order models. The maximum adsorption capacity equaled 189 mg g$^{-1}$. Satisfactory results from the application of AC for the elimination of 2,4-D from environmental waters (Algerian dam waters) synthetically polluted with 2,4-D were obtained (efficiency of 96%).

The same chemical activating agent (zinc chloride) was used to develop mushroom residues-based activated carbons for the removal of 2,4-D [89]. The differentiation in the pyrolysis temperature (400 and 500 °C) resulted in various values of $S_{BET}$ of the adsorbents (1067 and 799 m$^2$ g$^{-1}$) and of the adsorption capacity for the herbicide (105.4 and 118.2 mg g$^{-1}$, respectively). The maximum adsorption capacity of carbon pyrolyzed in higher temperatures equaled 241.7 mg g$^{-1}$ (pH = 4). The adsorption data were best described using the Langmuir and the linear driving force (LDF) models. It was shown that the use of the adsorbent resulted in a high degree of herbicide removal (70%) from the river water solution and, after the regeneration process, it was retained with high efficiency until the ninth cycle of reuse.

The elimination of 2,4-D and MCPA from an aqueous solution by activated carbons from rice straw in a batch system was investigated [90]. The ACs were prepared by chemical activation with $ZnCl_2$ and $H_3PO_4$, and their surface area was 771 and 613 m$^2$ g$^{-1}$, respectively. The adsorption kinetics was described well by the PSO model, while the adsorption isotherms were well fitted to the Langmuir model. The adsorption capacity of both the ACs for MCPA removal was greater than for 2,4-D.

Activated carbon from an oil palm trunk was fabricated via a chemical activation route using $H_3PO_4$ to test its ability to remove 2,4-D from aqueous solutions [91]. The effects of the pH, the temperature, the initial adsorbate concentration, and the contact time, on the

adsorption process were investigated. The highest removal efficiency was obtained at pH 2. The adsorption efficiency decreased as the temperature increased. The Langmuir isotherm model and the PSO kinetic model showed a better correlation with the experimental data. The monolayer maximum adsorption capacity was found to be 420.4 mg g$^{-1}$ at 25 °C.

The activated carbon prepared from peanut shells by chemical activation with $H_3PO_4$ was tested as an adsorbent for 2,4-D removal from water [92]. Adsorption experiments for 2,4-D removal were performed via a batch system, in which the effects of the initial 2,4-D concentration, the adsorbent dose, the contact time, the pH, and the temperature were studied. Moreover, the influence of the simultaneous presence of inorganic salts and humic acid on the adsorption process was also investigated. The kinetic results yielded well to the PSO model. The adsorption equilibrium was best fitted to the Langmuir model, and the maximum adsorption capacity of the AC for 2,4-D removal was 281.2 mg g$^{-1}$ at 30 °C.

Mesoporous activated carbon derived from *Peltophorum pterocarpum* pods was obtained by Samanth et al. [93] through chemical activation with $H_3PO_4$, followed by low-temperature carbonization. The surface area of the adsorbent was found to be 1078 m$^2$ g$^{-1}$ and its adsorption capacity for 2,4-D removal was 255.1 mg g$^{-1}$. The adsorption data were fitted using the Langmuir and pseudo-second-order models. The values on the adsorption energy and thermodynamic functions revealed physisorption and exothermicity.

Naboulsi et al. [94] generated activated carbon from *Ritama-Monosperma* (L.) *Boiss* powder through carbonization at 500 °C and activation with $H_3PO_4$. The influence of various experimental parameters (pH, temperature, adsorbate dose) on the equilibrium, kinetics, and thermodynamics of 2,4-D and 2,4,5-T adsorption from a bicomponent solution was evaluated. A maximum adsorption capacity of 69.44 mg g$^{-1}$ was found for 2,4-D and 13.81 mg g$^{-1}$ for 2,4,5-T. The Freundlich and the pseudo-second-order kinetic equations were indicated as working well for the theoretical description of the adsorption process.

A paper by Doczekalska et al. [95] reports the efficient removal of 2,4-D and MCPA during an aqueous phase, using activated carbons derived from various lignocellulosic materials. In that investigation, the ACs were prepared from willow, miscanthus, flax, and hemp shives, via KOH activation. The adsorption was pH-dependent, and the removal of both herbicides decreased with an increase in the initial pH of the solution. The adsorption kinetics of both herbicides was better represented by the PSO model, while the equilibrium data followed the Langmuir isotherm.

Trivedi et al. [45] proposed activated carbon synthesis from groundnut shells via chemical activation with KOH. The as-received AC was characterized by $S_{BET}$ = 709 m$^2$ g$^{-1}$ and an adsorption capacity for 2,4-D removal of 250 mg g$^{-1}$.

Activated carbon obtained by KOH activation of the biochar, obtained as a result of the pyrolysis of safflower press cake, was prepared and tested as an adsorbent for the removal of 2,4-D from aqueous solutions [96]. The effects of the initial pH value, the adsorbent dosage, the initial concentration, the contact time, and the temperature were evaluated. The results showed that the PSO model was more compatible with the experimental kinetic data and that the Langmuir isotherm model provided a better correlation for the adsorption of 2,4-D. The maximum adsorption capacity was found to be 344.8 mg g$^{-1}$ at 318 K. The thermodynamic studies revealed that the adsorption process was spontaneous and endothermic.

Koyuncu et al. [97], Rambabu et al. [98], and Pandiarajan et al. [99] prepared activated carbons from capsicum industrial processing pulp, date palm coir, and orange peel, respectively, via single-step KOH-catalyzed pyrolysis. The first research group demonstrated that the isotherm data were described well by the Langmuir isotherm model, and the maximum adsorption capacity for 2,4-D removal was 385 mg g$^{-1}$. The date palm coir-based activated carbon [98] was characterized by a flaky morphology, a well-graphitized porous structure (947 m$^2$ g$^{-1}$), and oxygen-rich surface moieties. The Langmuir and the pseudo-second-order models best described the equilibrium (with $q_m$ = 50.3 mg g$^{-1}$) and the kinetics of the adsorption process. An evaluation of the costs of the adsorbent production and regeneration was made, and cyclic reusability tests were also conducted. Pandiarajan's

group extended an adsorption study to other chlorophenoxy acid herbicides (2,4-DP, MCPA, 2,4,5-T, MCPP). The Langmuir model fitted all the herbicide adsorption isotherms very well. The results showed that the adsorption increased in the order: 2,4-DP < 2,4,5-T < MCPA < 2,4-D < MCPP. The pseudo-second-order kinetic model was indicated as the one with the highest fitting quality to the experimental data for all the adsorption systems.

Tefera and Tulu [100] tested activated carbon prepared from wheat straw treated with sulfuric acid and potassium hydroxide for the removal of 2,4-D from aqueous solutions. The removal efficiency of 2,4-D was evaluated using varying parameters, such as the adsorbent dose, temperature, pH, and contact time. It was observed that the adsorption efficiency of the herbicide increased with the increasing temperature, contact time, and adsorbent dose, and decreased with an increase in the solution pH. The equilibrium data were analyzed using the Langmuir and Freundlich isotherms and the Langmuir model best described the experimental data.

Herrera-García et al. [101] prepared activated carbon from agricultural waste made of yam peel (*Dioscorea rotundata*) by chemical activation with KOH. The as-prepared AC was chemically modified using magnetite nanoparticles. Both the ACs were characterized and tested as adsorbents for the removal of 2,4-D from aqueous solutions. The effects of the solution pH, the initial concentrations of 2,4-D, and the temperature, were investigated. The PFO, PSO, Elovich, and Webber–Morris kinetic equations, as well as the Langmuir, Freundlich, and Elovich isotherms, were used for the description of the experimental data. The results showed that the adsorption kinetics and isotherms were better described through the PSO and Freundlich models, respectively. Langmuir's maximum adsorption capacity values at 25 °C were 99.0 and 86.2 mg g$^{-1}$ for non-modified and $Fe_3O_4$-modified ACs, respectively.

Salman in the paper [102] dealt with the optimization of the conditions of activated carbon preparation from palm oil fronds using the physiochemical activation method (KOH and $CO_2$ as activating agents), and its application for the removal of various types of pesticides (e.g., bentazon, carbofuran, and 2,4-D). Of the variable synthesis parameters (activation time and chemical impregnation ratios KOH/biochar, activation temperature), the latter one was found to have the greatest effect on the activated carbon yield in regard to pesticides adsorption. The surface area of the activated carbon, prepared under optimum conditions, equaled 1237 m$^2$ g$^{-1}$. The optimizing analysis of variance was carried out using the response surface methodology (RSM).

Table 4 presents a compilation of the results of various activated carbons in the removal of 2,4-D and MCPA and their adsorption capacity.

The collected adsorption results showed that activated carbons derived from agricultural and household waste were very good adsorbents for the removal of chlorophenoxy herbicides from an aqueous solution. The adsorption capacities of activated carbons towards 2,4-D and MCPA were in the range of 1.015–592.7 mg g$^{-1}$ and 85–546.7 mg g$^{-1}$, respectively. The observed differences resulted mainly from the degree of development of the porous structure of the solids, the chemical characteristics of the surface, and the experimental conditions of the adsorption process. The high surface area of activated carbon, the basic nature of its surface, and the acidic conditions of the adsorption experiment favored chlorophenoxy herbicides adsorption. In general, the adsorption capacities of the activated carbons were very high when compared with other waste-based adsorbents, i.e., unmodified/modified materials, ashes, and biochars. Additionally, their adsorption capacities were not inferior to commercial activated carbons derived from fossil coals, for which the values of the parameter q$_m$ towards 2,4-D and MCPA were in the range of 8.11–555.5 mg g$^{-1}$ and 105.3–599.9 mg g$^{-1}$, respectively [33]. A significant share of micropores in the pores' structure caused the overall adsorption rate to be pore volume diffusion-dependent.

**Table 4.** The results from chlorophenoxy herbicides adsorption studies on activated carbons produced from agricultural and household waste.

| Adsorbent/ Activating Agent | $S_{BET}$ m² g⁻¹ | Adsorption Capacity ($q_m$), mg g⁻¹ | | Isotherm Model | Kinetic Model | Integral Cost–Benefit | Ref. |
|---|---|---|---|---|---|---|---|
| | | **2,4-D** | **MCPA** | | | | |
| AC from tea waste/ST | 576 | 58.8 | - | L, F | PFO, PSO, E | Low | [56] |
| AC from *Pinus taeda* wood/self-activation | 1450 | 471.7 | - | L, F, Te | PFO, PSO, E | High | [65] |
| AC from *Lagenaria vulgaris* shell/ST | 665 | 333.3 | - | L, F, S | PFO, PSO | High | [76] |
| AC from sugarcane bagasse/ST | 547 | 153.9 | - | L, F, R-P, Te, DR | PFO, PSO, E | Medium | [77] |
| AC from coconut shell/ST | 991 | 233.0 | - | L, F, R-P, Te, DR | PFO, PSO, E | High | [77] |
| AC from endocarp from babassu coconut/ST | 1068 | 235.5 | - | L, F, R-P, Te, DR | PFO, PSO, E | High | [77] |
| AC from canola stalk/ST | 556.8 | 135.8 | - | - | PFO, PSO, E | Medium | [78] |
| AC from peanut shells/ST | 1240 | 549 | - | F | PFO | High | [79] |
| AC from *Tectona grandis* tree sawdust (Teak-7360)/CD | 787 | - | 212.7 | L, F | - | High | [80] |
| AC from *Tectona grandis* tree sawdust (Teak-7480)/CD | 910 | - | 313 | L, F | - | High | [80] |
| AC from *Njiliti* wood waste/CD | 603 | - | 85 | - | - | Low | [81] |
| AC from *Candeia* wood waste/CD | 838 | - | 295 | - | - | High | [81] |
| AC from *Tchitiotidi* wood waste/CD | 741 | - | 220 | - | - | High | [81] |
| AC from *Nuati* wood waste/CD | 801 | - | 245 | - | - | High | [81] |
| AC from pistachio shell/CD | 556 | 453.4 | 314 | GL | PFO, PSO, E, m-exp, MOE | High | [82] |
| AC from pistachio shell/ST; CD | 669 | | 288.9 | GL | PFO, PSO, E, m-exp, MOE | High | [82] |
| AC from pistachio shell/CD; MV | 685 | | 286.9 | GL | PFO, PSO, E, m-exp, MOE | High | [82] |
| AC from Soybean waste/MV | 1696 | 253.2 | - | L, F, Te | - | High | [83] |
| AC from sesame seed/ZnCl₂ | - | 166.7 | - | L, F, Te, DR | PFO, PSO, E | Medium | [84] |
| AC from olive-waste cake/ZnCl₂ | 1418 | 129.9 | - | L, F | PFO, PSO | Medium | [85] |
| AC from queen palm endocarp/ZnCl₂ | 782 | 367.8 | - | L, F, Te, DR | PFO, PSO | High | [86] |
| AC from orange pulp/ZnCl₂ | 1779 | 71.9 | - | L, F, Te, DR | PFO, PSO, W-M | Low | [87] |
| AC from *Elaeagnus angustifolia* seeds/ZnCl₂ | 1109 | 189 | - | L, F, Te | PFO, PSO | Medium | [88] |

**Table 4.** *Cont.*

| Adsorbent/ Activating Agent | $S_{BET}$ m$^2$ g$^{-1}$ | Adsorption Capacity ($q_m$), mg g$^{-1}$ | | Isotherm Model | Kinetic Model | Integral Cost–Benefit | Ref. |
|---|---|---|---|---|---|---|---|
| | | 2,4-D | MCPA | | | | |
| AC from mushroom residues (AC400)/ZnCl$_2$ | 1067 | 105.4 | - | L, F, R-P | LDF | Medium | [89] |
| AC from mushroom residues (AC500)/ZnCl$_2$ | 799 | 241.7 | - | L, F, R-P | LDF | High | [89] |
| AC from rice straw/ZnCl$_2$ | 771 | 280.7 | 329.0 | L, F, P-R | PFO, PSO | High | [90] |
| AC from rice straw/H$_3$PO$_4$ | 613 | 271.8 | 329.0 | L, F, P-R | PFO, PSO | High | [90] |
| AC from oil palm trunk/H$_3$PO$_4$ | 1657 | 420.4 | - | L,F | PFO, PSO | High | [91] |
| AC from peanut shell/H$_3$PO$_4$ | 458.2 | 281.2 | - | L, F, Te | PFO, PSO, W-M | High | [92] |
| AC from *P. pterocarpum* pods/H$_3$PO$_4$ | 1078 | 255.1 | - | L | PSO | High | [93] |
| AC from *R.-M. (L.) Boiss* powder/H$_3$PO$_4$ | - | 69.4 | - | F | PFO, PSO, IDM | Low | [94] |
| AC from willow/KOH | 1280 | 510.5 | 484.4 | L, F | PFO, PSO | High | [95] |
| AC from miscanthus/KOH | 1420 | 569.5 | 537.1 | L, F | PFO, PSO | High | [95] |
| AC from flax shives/KOH | 1587 | 592.7 | 546.7 | L, F | PFO, PSO | High | [95] |
| AC from hemp shives/KOH | 1324 | 540.5 | 493.5 | L, F | PFO, PSO | High | [95] |
| AC from groundnut shell/KOH | 709 | 250.0 | - | L, F, Te | PFO, PSO | High | [45] |
| AC from biochar/KOH | 1277 | 344.8 | - | L, F, Te, DR | PFO, PSO | High | [96] |
| AC from capsicum pulp/KOH | 1564 | 385 | - | L,F | - | High | [97] |
| AC from date palm coir/KOH | 947 | 53 | - | L, F, Te | E, PFO, PSO | Low | [98] |
| AC from orange peel/KOH | 592 | 515.5 | 416.7 | E, IDM, L, F, Te, DR | PFO, PSO | High | [99] |
| AC from wheat straw/H$_2$SO$_4$; KOH | - | 1.015 | - | L,F | Contact time | Low | [100] |
| AC from yam peels/KOH | 715 | 99.0 | - | L, F, E | PFO, PSO, E | Low | [101] |
| AC from yam peels/KOH; Fe$_3$O$_4$ | 325 | 86.2 | - | L, F, E | PFO, PSO, E | Low | [101] |

## 3. Mechanism of Adsorption

Adsorption of chlorophenoxy herbicides by the different adsorbents, based on agricultural and household waste, is the result of various specific and nonspecific interactions. The adsorption mechanism of the same compound may change depending on the different properties of the adsorbent. The same situation occurs, for example, with a change in pH because the pH of the solution affects both the degree of dissociation and ionization of the adsorbate molecules, as well as the charge that will be present on the surface of the adsorbent. Thus, depending on the pH of the solution, adsorption can occur via electrostatic interactions (attractive or repulsive) between the adsorbate and the adsorbent.

The most typical and important forces driving adsorption by biochars and activated carbons include π-π interactions, electrostatic attraction and repulsion, hydrophobic interactions, covalent bonding and hydrogen bonding interactions between the herbicide molecules and the adsorbent surface functional groups. A significant share of the micro- or small mesopores in the structure of these materials causes the adsorption to proceed mainly through the pore-filling mechanism. Figure 1a presents possible supramolecular systems created during the adsorption of the 2,4-D molecule on the exemplary biochar [66]. Impregnation of the raw material with $H_3PO_4$ at the pre-treatment stage of synthesis determined the occurrence of typical interactions between the adsorbate and the adsorbent, as well as specific ones resulting from the activating agent used (i.e., with metaphosphate groups). The high value of the maximum adsorption capacity for this biochar towards 2,4-D (323.76 mg g$^{-1}$) at pH = 2 and T = 318 K was found. Additionally, the morphological structure of the used adsorbent and the herbicide adsorption isotherms are illustrated in Figure 1b–d, respectively.

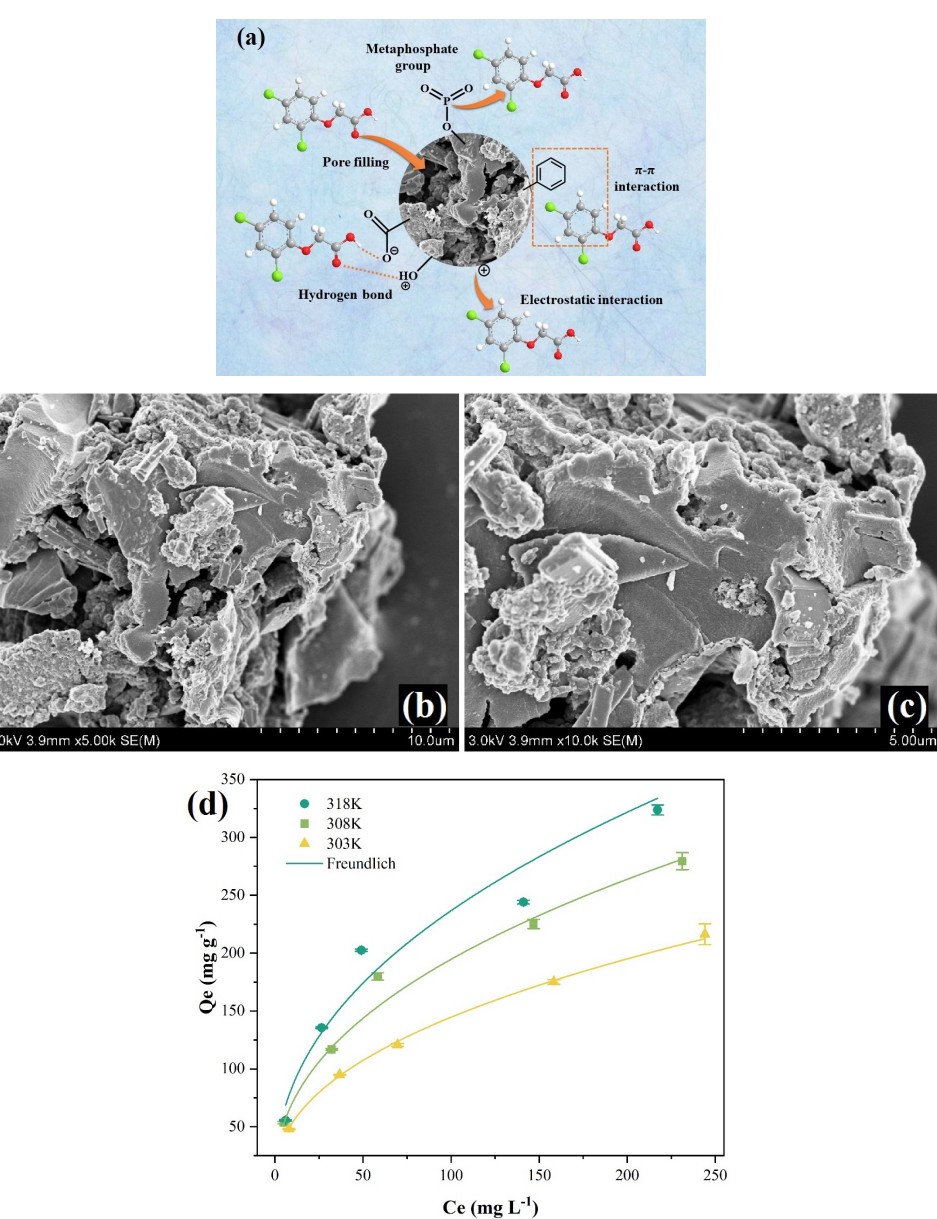

**Figure 1.** The possible interactions in the 2,4-D adsorptive process using coffee ground char (**a**). SEM images of the adsorbent (**b**,**c**). The herbicide adsorption isotherms on the biochar (**d**) Reprinted with permission from Ref. [66]. Copyright 2023, copyright Elsevier.

Regarding unmodified and modified natural materials and ashes, the mechanism of the 2,4-D or MCPA adsorption must be considered individually in conjunction with the unique (individual) physicochemical properties of the adsorbent. According to the authors of some works [35–37,39], the adsorption of chlorophenoxy herbicides by rice husks with a positive surface charge, follows a chemical ion-exchange mechanism or occurs via a chemical process (involving valency force sharing). Chemical reactions (through free electron pairs of surface functional groups, e.g., carbonyl, amide, ether), cation–dipole interactions (due to the occurrence of hydroxyl, amine, and amide groups), as well as hydrogen bonds are proposed for the adsorption of 2,4-D by saw-sedge Cladium mariscus (*C. mariscus*). In turn, for bio-waste materials, such as apple shell, orange peel, banana peel, and millet residue, the discussed process results from molecular interactions (when the molecular form of the pesticide occurs) and repulsive electrostatic interactions (at a solution pH where both the adsorbent surface moieties, i.e., carboxyl and sulphate, and 2,4-D are negatively charged) [40]. Al-Zaben and Mekhamer [41] reported no relationship between the MCPA removal efficiency and the pH solution using coffee waste as the adsorbent. The existence of an electrostatic interaction was excluded in this system, and the responsibility for the adsorption was assigned to the hydrogen bonds between the OH moieties on the adsorbent surface and the acidic group of the adsorbate molecules.

According to the authors of papers [47–49], the oxidation of various biomass (*Physalis peruviana* fruit, *Arachis hypogaea* skins and wheat husks) with sulfuric acid resulted in oxygenated functionalities on their surface, which participated in the adsorption of the chlorophenoxy herbicides via electrostatic forces, hydrogen bonding, and van der Waals forces. The authors of the latter paper [49] also indicated halogen bonds and $\pi$–$\pi$ interactions as responsible for adsorption. Noteworthy is the relatively high adsorption efficiency of acid-modified wheat husks, with the capacity of 161.1 mg g$^{-1}$ (pH = 2, T = 298 K). For this reason, the possible supramolecular systems of this adsorbent with the 2,4-D molecule (Figure 2a), along with the morphological structure of the adsorbent (Figure 2b–d), and the equilibrium isotherms at various experimental temperatures (Figure 2e), are presented.

The modification of tiger nut residue using N-cetylpyridinium [50] resulted in obtaining the adsorbent, which interacted with 2,4-D through electrostatic attraction, hydrogen bonding, and triggered $\pi$-$\pi$ stacking. Trivedi's comprehensive research [42–46] on ashes originating from various waste materials showed a close relationship between their chemical composition and the mechanism of the chlorophenoxy herbicides adsorption. The presence of metallic oxides (CaO, K$_2$O) in ash, which in water take the form of metallic hydroxides, resulted in electrostatic repulsion forces between them and adsorbate anions. In turn, for Al$_2$O$_3$, which constitutes positive centers on the adsorbent surface, the triggering of attractive electrostatic interactions was observed. Thus, the mineral composition of ash determines the mechanism and effectiveness of the adsorption process. For hydrochars (HCs), the mechanism of the 2,4-D adsorption via intensive partitioning and/or chemisorption was postulated [64]. Regarding the interactions of pollutants with more advanced magnetic waste-based materials [52,53] with various functional groups (hydroxyl, amine, carboxyl), the hydrogen bonding, electrostatic complexation, and $\pi$-$\pi$ mechanisms were suggested. Moreover, some of these interactions were additionally enhanced by the occurrence of iron. The high efficiency of the ferrospinel composite (FLPWAC) [53] ($q_m$ = 400 mg g$^{-1}$, T = 318 K) was found. The possible supramolecular adsorbate–adsorbent systems (Figure 3a), the morphological structure of the adsorbent (Figure 3b–f), and the equilibrium isotherms at various experimental temperatures (Figure 3g), are presented.

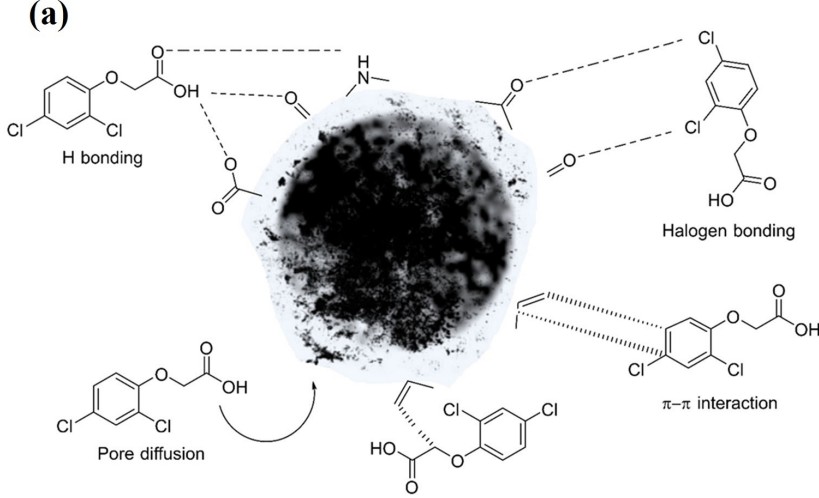

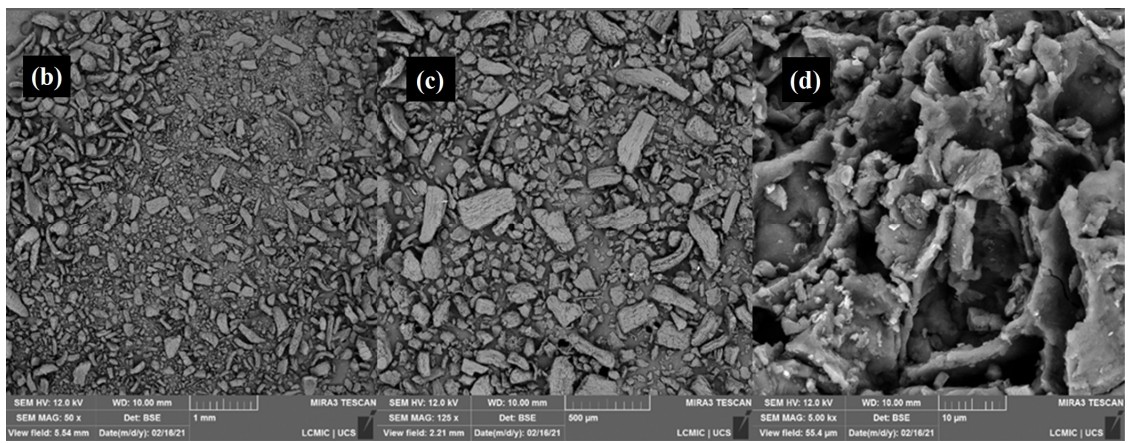

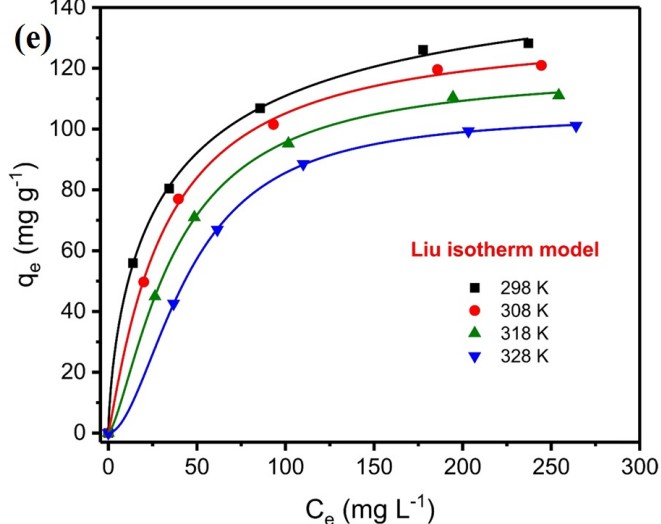

**Figure 2.** The possible interactions in the 2,4-D adsorptive process using acid-modified wheat husks (**a**). SEM images of the adsorbent (magnitudes: (**b**) ×500, (**c**) ×1000, (**d**) ×3000), (**b–d**). Equilibrium isotherms of the 2,4-D adsorption by modified wheat husks (pH = 2) (**e**) Reprinted with permission from Ref. [49]. Copyright 2021, copyright Elsevier.

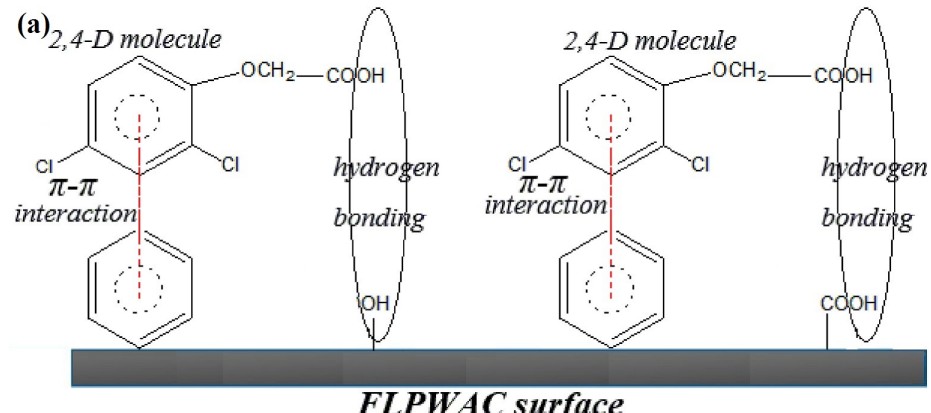

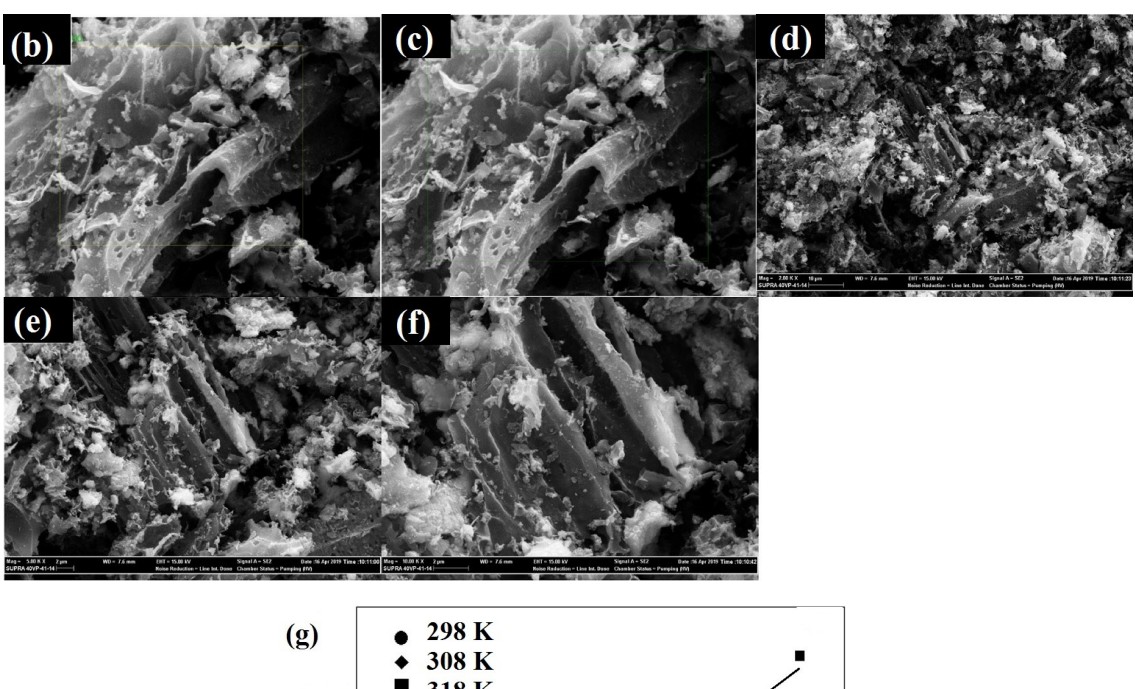

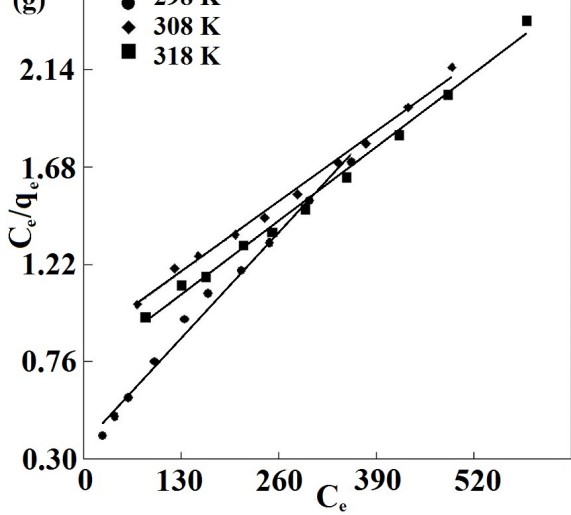

**Figure 3.** The possible interactions in the 2,4-D adsorptive process using the ferrospinel composite (**a**). SEM images of the adsorbent at different expansion ratios (**b–f**). Langmuir plots of the 2,4-D adsorption on the composite (**g**). Reprinted with permission from Ref. [53]. Copyright 2020, copyright Elsevier.

Summing up, the variety of types of adsorbents and their different characteristics translate into a multitude of mechanisms leading to the adsorption of chlorophenoxy herbicides.

## 4. Effect of Co-Substances on Chlorophenoxy Herbicides Adsorption by Waste-Based Adsorbents

Environmental waters are complex biological systems, containing several compounds of natural and anthropogenic origin, so research on the assessment of the impact of various co-substances on chlorophenoxy herbicide removal by waste-based adsorbents is enormously important. This issue has been addressed in only a few papers out of all the ones cited. The studies concerned the co-existence of mineral salts, organic matter, dyes, and other herbicides, but there was no information on the co-existence of adjuvants, which are auxiliary substances in commercial formulations of herbicide preparations.

Fatombi et al. [92] and Almahri [67] reported that 2,4-D removal efficiency by activated carbon and biochar in the presence of NaCl, $CaCl_2$, and humic acid (HA) decreased, wherein the influence of $CaCl_2$ was lower than NaCl. Such a difference was explained by the higher polarizability effect of the $Ca^{2+}$ ion and the lower thickness of the double layer in the solid with the participation of this ion. The adsorption capacities were found to be 281.22, 86.12, 160.32, and 154.96 mg g$^{-1}$ for adsorption of 2,4-D only, and with the co-existence of NaCl, $CaCl_2$, and HA, respectively. It was also noticed that an increase in the salt concentration negatively affected herbicide adsorption, due to a higher dissociation degree of the 2,4-D molecules and the forming of pairs between the salt cations and herbicide anions. The slighter impact of $CaCl_2$ on 2,4-D adsorption was related to the screening effect of the negative charge of 2,4-D by the formation of complexes based on intermolecular bridges (R–COO—-$Ca^{2+}$—-OOC–R).

A similar trend in the influence of the presence of natural organic matter on 2,4-D adsorption by AC was also noticed by Gurav's research group [65]. Here, the adsorption capacities equaled 491.76 and 466.73 mg g$^{-1}$ for 2,4-D adsorbed from single- and bicomponent solutions, respectively. However, the effect of salinity on 2,4-D removal was reversed compared to that observed by Fatombi et al. [92]. The occurrence of NaCl in water in a certain concentration range (1–4 g L$^{-1}$) slightly enhanced the 2,4-D adsorption (from 493.86 to 496.39 mg g$^{-1}$), but a further increase in salt concentration caused a decrease in adsorption (to 489.08 mg g$^{-1}$). The enhancement of the adsorption with the rise in the ionic strength of the solution was connected with the screening effect of the carbon surface charge by NaCl.

In the paper [82], the effect of competing substances was analyzed for MCPA adsorption by two activated carbons, differentiated in terms of their porous structure and surface functionality. A slight effect of crystal violet co-existence on the MCPA adsorption was revealed, which was explained by the affinity of both pollutants to different adsorption sites. The large dye molecules were adsorbed in the mesopores, while the small MCPA ones were mainly located in the micropores. Thus, the authors stated that the adsorption of both compounds was mostly independent. However, a significant influence of the competing substance was observed when in the solution there were two substances belonging to the same group of herbicides (MCPA and 2,4-D). Here, a decrease in MCPA adsorption with the co-existing substance was a consequence of the occupation of the same adsorption sites ($q_m$ equals 314 and 144.43 mg g$^{-1}$ for the MCPA adsorbed from single- and bicomponent solutions, respectively). Relatively similar physicochemical properties and structures of the herbicides caused strong competition between each other in the adsorption process by activated carbon. It was also noticed that 2,4-D showed greater affinity to the carbon surface than MCPA due to the greater adsorption strength, which favored keeping 2,4-D in the adsorption active sites or replacing it after MCPA desorption. The adsorption capacities were found to be 242.58 and 144.43 mg g$^{-1}$ for 2,4-D and MCPA adsorbed from the bicomponent solution, respectively.

Naboulsi et al. [94] studied the competition effect on the adsorption of 2,4-D and 2,4,5-T by waste-based ACs. The maximum adsorption capacity of 69.44 mg g$^{-1}$ for 2,4-D and 13.81 mg g$^{-1}$ for 2,4,5-T for the bicomponent system was determined. The application of density functional theory (DFT) enabled the identification of 2,4-D as a more selective

pollutant due to the greater reactivity of its molecules (more active sites, either nucleophilic or electrophilic in nature).

A higher selectivity of the biomass-MOF composite towards 2,4-D compared to tetracycline (antibiotic) in binary systems was also observed [52]. Such behavior by the adsorbent was attributed to the relatively smaller size of the herbicide, which reduced the steric hindrance. However, it was found that an increase in the adsorbent dose reduced the adsorption efficiency due to grains agglomeration (limiting accessibility to the surface area).

In the paper [69], promising results from 2,4-D adsorption studies on biochar in multi-component systems were presented. A 100% biochar performance for pollutants from drainage water and 96.8% from simulated one-component wastewater was noted. This was due to the synergistic effect of the coexisting cations and anions in the drainage water on herbicide adsorption.

The 2,4-D removal in the range of 86.5–95.6% by biochar from real water (river effluent, drinking water, and wastewater) was achieved by Almahri et al. [67]. Only slightly worse results on 2,4-D removal efficiencies (72–76.30%) by sulfuric acid-treated peanut skins and wheat husks were found in the treatment of synthetic wastewater, consisting of a mixture of 2,4-D, atrazine, NaCl, KCl, or the simulated effluent from two Brazilian rivers [48,49].

The studies on various co-existing substances and their impact on the chlorophenoxy herbicides adsorption by waste-based adsorbents showed the complexity of the interactions between the components of adsorption systems and highlighted the significance of taking them into account in the development and optimization of new adsorbents for environmental applications. However, taking into account the complex composition of natural water for a better understanding of the interactions between other co-substances, herbicides, and adsorbents, further research in this direction is required.

## 5. Regeneration and Reusability of Waste-Based Activated Carbons

The evaluation of the recycling and reuse potential of waste-based ACs is another crucial factor in determining the implementation of these materials for adsorption applications. This is related to cost reduction for new adsorbents and actions to prevent possible secondary pollution of the environment with herbicides and other co-adsorbates, as a result of their leaching due to environmental factors from used activated carbon deposits at storage sites. The proper management of spent adsorbents seems to be such an important issue that many research groups have initiated studies in this direction. Generally, chemical, thermal, physical, and biological methods are used to regenerate activated carbons. Regarding the regeneration of waste-based activated carbons with adsorbed chlorophenoxy herbicides, the commonly used chemical method was based on the desorption of the adsorbate using NaOH solution as an eluent [50,65,84,86,89,92,93,98], and less frequently using other specific solvents or solutions, such as acetone [78], ethanol [78], water [78,98], and NaCl. The performance of this method depended on the interactions between the herbicide–activated carbon and between the herbicide–regenerating chemical agent. The adsorption efficiency of the regenerated adsorbent usually decreased with the number of adsorption–desorption cycles (maximum number of cycles 3–5), but some adsorbents could be reused even 7–9 times, with the maintenance of the adsorption effectiveness close to the initial use [65,86,89,98]. The decrease in the adsorption capacity in the following regeneration stages resulted from various factors: (i) the decrease in the surface sites' adsorption potential, (ii) steric effects, (iii) the disintegration of the pore structure, and (iv) the inability of the herbicide to desorb from the carbon structure. Of the solvents, e.g., acetone, ethanol, and water, the first one appeared to be the superior agent for AC regeneration. The thermal method of regeneration was applied by Bojić et al. [76] and Herrera-García et al. [101], with 49–60% keeping their initial adsorption capacities after five cycles. More satisfying results (70–79% initial efficiency) were obtained for the physical method of regeneration using a microwave heating oven. All these findings showed that ACs successfully passed the regeneration and reusability tests. However, for the application of adsorbents on an industrial scale, it is advisable to perform additional adsorption studies within a column or

a continuous flow system (fluidized-bed column) for the treatment of water polluted with chlorophenoxy herbicides.

Research in the field of regeneration and reusability of adsorbents makes economic and practical sense if a given material shows high adsorption capacity towards pollutants. Therefore, this issue has been described in only a few papers devoted to materials other than activated carbon. More precisely, they concerned coffee waste-based biochar and acid-modified wheat husk, with adsorption capacities for 2,4-D of 276.3 and 161.1 mg g$^{-1}$, respectively [49,67]. As in the case of many active carbons, the best results (retaining 45–84% removal efficiency after five cycles of adsorption–desorption) were obtained using sodium hydroxide as a desorbing agent. Nitric acid, calcium chloride, hydrochloric acid, ethylenediamine tetraacetic acid (EDTA), and ultrasonic cleaning with alcohol, were found not to be effective enough in the desorption of pollutants from biochars [67,70]. There were also materials that were treated with NaOH after use, but their subsequent efficiency was negligible. An example of such a material is a peanut husk-based MOF composite with magnetic properties [52]. Nevertheless, the regeneration process used allowed the recovery and reuse of the adsorbate, while the adsorbent disposed into soil acted as organic amendment due to the slow release of its major elements (i.e., Fe, C, N, and O).

## 6. Cost of Production of Waste-Based Activated Carbons

A review of the relevant literature revealed the selectivity and satisfying adsorption capability of waste-based activated carbons towards chlorophenoxy herbicides in model experiments both in single-component and multi-component systems. Importantly, these adsorbents could be regenerated using simple methods, which enables multiple use at a high level of efficiency. Another issue to analyze is the cost effectiveness of waste-based AC production compared to commercial materials. In a paper from 2022 [97], the authors estimated the production cost of a Capsicum pulp-based AC at approximately USD 1.75 per kg (summing up the costs of the raw material transport; the reagents, namely nitrogen gas, KOH, and hydrochloric acid; and the media, namely power, water), which is lower than that for commercial ACs, e.g., USD 2–5 per kg. In other works from 2021, the fabrication of ACs from mushroom residues [89] and date palm coir waste [98] was assessed to be USD 2.39 per kg (the total costs of reagents, namely nitrogen gas, hydrochloric acid, and zinc chloride; and media, namely power) and USD 3 per kg (the total costs of the capital expenses, namely the installation equipment, land, buildings, service; and the operating ones, namely the precursor collection, water, electricity, and labor), respectively. According to [98], the cost of commercial activated carbon may even reach USD 15 per kg. Actually, only a few cited works contained cost estimations, but based on the available data, the undertaking to expand the production of these adsorbents from a laboratory scale to an industrial one seems to be economically viable. Moreover, this project is in line with the trend of sustainable waste management, which is beneficial for the natural environment.

## 7. Conclusions

This paper presents a wide range of agricultural and household wastes as low-cost alternative adsorbents for the removal of two common herbicides, 2,4-D and MCPA, from water. The characteristics of the adsorbents, the conditions of the adsorption experiments and their results, and the theoretical models that best describe the adsorption process have been reported. Based on the available literature, mainly from the last decade, the use of waste materials as low-cost adsorbents, both in their original form and after modification, mainly in the form of waste-based biochar and activated carbon, was shown to be very attractive. Such adsorbents are not only effective, but are also very interesting from an economic point of view. The use of waste itself as an adsorbent reduces the cost of waste disposal and, thus, promotes environmental protection, and the low cost of these materials means that they can be disposed of after use without costly regeneration. However, the problems concerning the regeneration and reusability of waste-based activated carbons and the costs involved have been discussed. The waste materials presented in this paper

showed a satisfactory affinity with 2,4-D and MCPA, and since they are relatively low- or no-cost, readily available, and easily modified, e.g., by pyrolysis and activation, their use as alternative adsorbents seems very interesting and promising. The development of adsorbents using agricultural and household waste offers an interesting alternative to commercial materials for environmental applications and at the same time offers a way forward for proper waste management.

At the same time, it should be added that the presented review on the use of adsorbents based on waste materials for water purification, also showed the need to intensify the research towards extending it to more complex systems corresponding to real ones. Only a few studies concern this problem, while water and sewage contain many toxic substances with very diverse properties, originating from various sources, including those used as pesticide additives. The aim of adsorption techniques is to remove substances to the highest possible extent, which is difficult to achieve in such complex systems, in which a large part of the components may compete for access to the adsorption centers. Therefore, an important direction of future research is the analysis of adsorption in model and real multi-component systems. Taking into account the diversity of the physicochemical properties of the toxic substances present in water and sewage, including the molecular size, hydrophobic or hydrophilic nature, and molecular form, it is also necessary to analyze materials or systems of materials with a different porosity and chemical nature of the surface in order to achieve the maximum possible level of purification. On the other hand, in many cases the selective adsorbents should be applied; thus, also the studies concerning the mutual interactions between the adsorbates in complex systems, which can decrease the adsorption effectiveness of a given substance, are of great importance.

**Author Contributions:** Conceptualization, M.B., K.K., A.S. and A.D.-M.; data curation, M.B. and K.K.; visualization, M.B. and K.K.; formal analysis, K.K., A.S. and A.D.-M.; supervision, K.K., A.S. and A.D.-M.; writing—original draft, M.B., K.K. and A.S.; writing—review and editing, K.K., A.S. and A.D.-M. All authors have read and agreed to the published version of the manuscript.

**Funding:** This research received no external funding.

**Institutional Review Board Statement:** Not applicable.

**Informed Consent Statement:** Informed consent was obtained from all subjects involved in the study.

**Data Availability Statement:** The data are available from the corresponding author.

**Conflicts of Interest:** The authors declare no conflict of interest.

## Abbreviations

| | |
|---|---|
| AC | Activated carbon |
| A | Avrami kinetic model |
| Ba | Bangham kinetic model |
| DR | Dubinin–Radushkevich isotherm |
| E | Elovich isotherm model |
| F | Freundlich isotherm model |
| GL | The generalized Langmuir isotherm |
| Hi | Hill isotherm model |
| IDM | Intraparticle diffusion model |
| ITM | Theoretical adsorption model |
| J | Jovanovic equilibrium model |
| K | Khan equilibrium model |
| K-C | Koble–Corrigan equilibrium model |
| $k_i$ | Intraparticle diffusion rate in W-M kinetic model, $\frac{mg}{g}\sqrt{\frac{1}{min}}$ |
| $k_1$ | The rate constant for the PFO kinetic model, $min^{-1}$ |
| $k_2$ | The rate constant for the PSO kinetic model, $g\,mg^{-1}\,min^{-1}$ |

| L | Langmuir isotherm |
| LDF | Linear driving force kinetic model |
| L-F | Langmuir–Freundlich isotherm |
| m-exp | Multi-exponential kinetic equation |
| MOE | Mixed-order equation |
| PFO | Pseudo-first-order kinetic model/equation (PFOE) |
| PSO | Pseudo-second-order kinetic model/equation (PSOE) |
| $q_m$ | The maximum adsorption capacity, mg g$^{-1}$ |
| R-P | Redlich–Peterson isotherm |
| S | Sips isotherm |
| Te | Temkin isotherm |
| To | Toth isotherm |
| W-M | Weber–Morris kinetic model |

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
