# Peer review of "Waste-Based Adsorbents for the Removal of Phenoxyacetic Herbicides from Water: A Comprehensive Review"

_sustainability, doi:10.3390/su152316516_

Round 1
Reviewer 1 Report
Comments and Suggestions for Authors
The article provides an exhaustive review of adsorbents derived from agricultural and household waste for the removal of chlorophenoxy herbicides from water. It deals with various aspects, including the classification of adsorbents, the modification methods used, adsorbent characteristics, adsorption measurement conditions, and theoretical modeling. It also identifies the most effective adsorbents based on their capacity and rate of adsorption, addressing economic and ecological considerations.
However, the primary drawback of this paper lies in its lack of novelty. The use of waste materials for adsorption, especially in the context of chlorophenoxy herbicide removal, is a well-established and extensively researched area. The article does not introduce any innovative or groundbreaking ideas, methods, or materials that significantly advance the existing body of knowledge. Instead, it essentially reiterates what has been widely documented in the literature.
Furthermore, the article does not offer any clear future perspectives. Readers are left without guidance on potential areas of further research, emerging trends, or unexplored avenues in the field of herbicide removal. It would be beneficial to include a forward-looking section that identifies knowledge gaps and suggests potential directions for future studies, encouraging readers to envision the possibilities for further advancement.
While the article provides a comprehensive overview of adsorbents for chlorophenoxy herbicide removal, it falls short in terms of novelty and lacks a forward-looking perspective, potentially limiting its appeal to readers seeking fresh insights and research directions in this area.
Comments on the Quality of English LanguageMinor editing of the English language is required.
Author Response
Manuscript Number: Sustainability-2718239
Title: Adsorbents Based on Agricultural and Household Waste for the Removal of Phenoxyacetic Herbicides from Water: a Comprehensive Review
Reviewer #1: Comments on: Sustainability-2718239
The article provides an exhaustive review of adsorbents derived from agricultural and household waste for the removal of chlorophenoxy herbicides from water. It deals with various aspects, including the classification of adsorbents, the modification methods used, adsorbent characteristics, adsorption measurement conditions, and theoretical modeling. It also identifies the most effective adsorbents based on their capacity and rate of adsorption, addressing economic and ecological considerations.
However, the primary drawback of this paper lies in its lack of novelty. The use of waste materials for adsorption, especially in the context of chlorophenoxy herbicide removal, is a well-established and extensively researched area. The article does not introduce any innovative or groundbreaking ideas, methods, or materials that significantly advance the existing body of knowledge. Instead, it essentially reiterates what has been widely documented in the literature.
Furthermore, the article does not offer any clear future perspectives. Readers are left without guidance on potential areas of further research, emerging trends, or unexplored avenues in the field of herbicide removal. It would be beneficial to include a forward-looking section that identifies knowledge gaps and suggests potential directions for future studies, encouraging readers to envision the possibilities for further advancement.
While the article provides a comprehensive overview of adsorbents for chlorophenoxy herbicide removal, it falls short in terms of novelty and lacks a forward-looking perspective, potentially limiting its appeal to readers seeking fresh insights and research directions in this area.
Comments on the Quality of English Language
Minor editing of the English language is required.
Responses to Reviewer 1 suggestions:
The article presents a wide range of agricultural and household wastes as low-cost alternative adsorbents for the removal of two commonly used herbicides (2,4-D and MCPA) from water. The characteristics of the adsorbents, the conditions for adsorption experiments and their results, and the theoretical models that best describe the adsorption process have been described. Based on recent literature, the suitability and effectiveness of waste materials as low-cost adsorbents, both in their original form and after modification, mainly to waste biocarbon and activated carbon, have been demonstrated. The following chapters discuss the effect of co-substances on the adsorption of chlorophenoxy herbicides on waste-based adsorbents, as well as the problems of regeneration and reusability of waste-based activated carbons and the associated costs. The waste materials presented in this work have shown a satisfactory affinity for 2,4-D and MCPA, and since they are relatively cheap (or completely free), readily available, and easily modified, e.g. by pyrolysis and activation, their use as alternative adsorbents seems very interesting and promising. The use of the waste itself as an adsorbent reduces the cost of waste disposal and thus promotes environmental protection, and the low cost of these materials means that they can be disposed of after use without costly regeneration.
In our work, we have clearly demonstrated that the development of adsorbents using agricultural and household wastes is an interesting alternative to commercial materials for environmentally friendly applications, and at the same time a way to manage them properly. We indicated the potential areas for further research and emerging trends.
As for a novelty of this article, a large number of publications on the use of waste materials for adsorption, especially in the context of the removal of chlorophenoxy herbicides, indicates the topicality of the subject and the great interest of many research centers (and therefore the accessibility to a wide range of readers).
Such a large number of publications (and thus a large number of different adsorbents used) means that this knowledge is indeed, as the reviewer claims, well established and extensively researched, but unfortunately very scattered. In our review, we have set ourselves the task of bringing together the current state of knowledge on this subject in one place. In this respect, we believe that this is a very valuable and attractive project for potential readers. The presentation of the current state of knowledge and the comparison of many different adsorbents may help many researchers to take their research forward. Despite the great interest in the topic of adsorption of phenoxyacetic herbicides from water and a large number of published research articles, there are no valuable review papers that organize the current developments and the current state of knowledge. To the best of our knowledge, the reviewed paper would be only the second of its kind. The first review (and the only one to our knowledge) concerns the adsorption of phenoxyacetic herbicides from water on carbonaceous and non-carbonaceous adsorbents (Molecules 28 (2023) 5404. https://doi.org/10.3390/molecules28145404).
Regarding the Reviewer suggestions the manuscript was enriched by adding the section “3. Mechanism of adsorption”, and some remarks concerning the economic and ecological aspects. Moreover, the Conclusions were widened by underlying the perspective trends.
“3. Mechanism of adsorption
Adsorption of chlorophenoxy herbicides on the different adsorbents based on agricultural and household waste is the result of various specific and nonspecific interactions. The adsorption mechanism of the same compound may change depending on the different properties of the adsorbent. The same situation occurs, for example, with a change in pH because the pH of the solution affects both the degree of dissociation and ionization of the adsorbate molecules, as well as the charge that will be present on the surface of the adsorbent. Thus, depending on the pH of the solution, adsorption can occur via electrostatic interactions (attractive or repulsive) between the adsorbate and the adsorbent.
The most typical and important forces driving adsorption on biochars and activated carbons include π-π interactions, electrostatic attraction and repulsion, hydrophobic interactions, covalent bonding and hydrogen bonding interactions between herbicide molecules and adsorbent surface functional groups. A significant share of micro- or small mesopores in the structure of these materials causes that adsorption proceeds mainly by the pore-filling mechanism. Figure 1a presents possible supramolecular systems created during adsorption of 2,4-D molecule on the exemplary biochar [66]. Impregnation of raw material with H3PO4 at the pre-treatment stage of synthesis determined the occurrence of typical interactions between the adsorbate and the adsorbent, as well as specific ones resulted from activating agent used (i.e. with metaphosphate groups). The great value of maximum adsorption capacity for this biochar towards 2,4-D (323.76 mg g−1) at pH = 2 and T = 318 K was found. Additionally, the morphological structure of used adsorbent and the herbicide adsorption isotherms are illustrated in Figures 1b-c and 1d, respectively.
Figure 1. The possible interactions in the 2,4-D adsorptive process using coffee ground char (a). SEM images of the adsorbent (b-c). The herbicide adsorption isotherms on biochar (d). [66].
Regarding unmodified, modified natural materials and ashes, mechanism of the 2,4-D or MCPA adsorption must be considered individually in conjunction with unique (individual) physicochemical properties of the adsorbent. According to the authors of some works [35-37,39] adsorption of chlorophenoxy herbicides on rice husk with positive surface charge, follows a chemical ion-exchange mechanism or via a chemical process (involving valency force sharing). Chemical reactions (through free electron pairs of surface functional groups e.g., carbonyl, amide, ether), cation–dipole interactions (due to the occurrence of hydroxyl, amine, and amide groups) as well as hydrogen bonds are proposed for adsorption of 2,4-D on the saw-sedge (C. mariscus). In turn, for bio-waste materials such as apple shell, orange peel, banana peel, and millet residue, the discussed process results of molecular interactions (when the molecular form of the pesticide occurs) and repulsive electrostatic interactions (at solution pH where both the adsorbent surface moieties i.e. carboxyl and sulphate, and 2,4-D are negatively charged) [40]. Al-Zaben and Mekhamer [41] reported no relationship between the MCPA removal efficiency and the pH solution for coffee waste as adsorbent. The existence of electrostatic interaction was excluded in this system, and the responsibility for adsorption was assigned to hydrogen bonds between OH moieties on the adsorbent surface and acidic group of adsorbate molecules.
According to authors of papers [47-49] oxidation of various biomass (Physalis peruviana fruit, Arachis hypogaea skins and wheat husks) with sulfuric acid resulted in oxygenated functionalities on their surface which participated in adsorption of the chlorophenoxy herbicides via electrostatic forces, hydrogen bonding and van der Waals forces. The authors of the latter paper [49] indicated also halogen bonds and π–π interactions as responsible for adsorption. Noteworthy is the relatively high adsorption efficiency of acid-modified wheat husks, with the capacity of 161.1 mg g−1 (pH = 2, T = 298 K). For this reason, the possible supramolecular systems of this adsorbent with the 2,4-D molecule (Figure 2a) along with the morphological structure of adsorbent (Figure 2b-d), and the equilibrium isotherms at various experimental temperatures (Figure 2e) are presented.
Figure 2. The possible interactions in the 2,4-D adsorptive process using acid-modified wheat husks (a). SEM images of adsorbent (magnitudes (b) × 500, (c) × 1000 (d) x 3000) (b-d). Equilibrium isotherms of the 2,4-D adsorption on modified wheat husks (pH = 2) (e) [49].
Modification of tiger nut residue using N-cetylpyridinium [50] resulted in obtaining the adsorbent, which interacts with 2,4-D through electrostatic attraction, hydrogen bonding and a triggered π-π stacking. Trivedi's comprehensive research [42-46] on ashes originated from various waste materials showed a close relationship between their chemical composition and the mechanism of the chlorophenoxy herbicides adsorption. The presence of metallic oxides (CaO, K2O) in ash, which in water take a form of metallic hydroxides resulted in electrostatic repulsion forces between them and adsorbate anions. In turn, for Al2O3 which constitutes positive centers on the adsorbent surface, the trigger attractive electrostatic interactions were observed. Thus, the mineral composition of ash determines mechanism and effectiveness of the adsorption process. For hydrochars (HCs) mechanism of the 2,4-D adsorption via intensive partitioning and/or chemisorption was postulated [64]. Regarding the interactions of pollutants with more advanced magnetic waste-based materials [52,53] with various functional groups (hydroxyl, amine, carboxyl), the hydrogen bonding, electrostatic complexation, and π-π mechanisms were suggested. Moreover, some of these interactions were additionally enhanced by the occurrence of iron. The high efficiency of ferrospinel composite (FLPWAC) [53] (qm = 400 mg g−1, T=318 K) was found. The possible supramolecular adsorbate-adsorbent systems (Figure 3a), the morphological structure of adsorbent (Figure 3b-f), and the equilibrium isotherms at various experimental temperatures (Figure 3g) are presented.
Figure 3. The possible interactions in the 2,4-D adsorptive process using the ferrospinel composite (a). SEM images of adsorbent at different expansion ratios (b-f). Langmuir plots of the 2,4-D adsorption on composite (g) [53].
Summing up, the variety of types of adsorbents and their different characteristics translate into a multitude of mechanisms leading to the adsorption of chlorophenoxy herbicides.”
“All these findings showed that ACs successfully passed the regeneration and reusability tests. However, for application of adsorbents on an industrial scale, it is advisable to perform additional adsorption studies within column or a continuous flow system (fluidized bed column) for treatment of water polluted with chlorophenoxy herbicides.” (5. Regeneration and reusability of waste-based activated carbons)
“Various unmodified, modified agricultural and household waste materials and ashes, and their adsorption performance in chlorophenoxy herbicides removal are summarized in Table 2. Additionally, the cost-benefit of the given adsorbents was evaluated based on the comparison of their adsorption capacity towards chlorophenoxy herbicides with synthesis cost, the impact of their production on the natural environment (energy consumption, harmfulness of reagents used, secondary waste) and their recyclability. Such subjective evaluation is widely accepted and practiced by many scientists [54,55] “ (2.1. Unmodified, modified natural materials and ashes)
The exemplary Table:
|
Adsorbent |
SBET m2 g−1 |
Adsorption capacity (qm) mg g−1 |
Isotherm model |
Kinetic model |
Integral cost - benefit |
Ref. |
|
|
2,4-D |
MCPA |
||||||
|
rice husk ash |
34 |
1.425 |
- |
L, F, Te |
PFO, PSO, E |
low |
[35] |
|
rice husk ash |
34 |
- |
1.681 |
L, F, Te |
PFO, PSO, E |
low |
[37] |
|
nanosized rice husk |
- |
24.75 |
- |
L, F, Te, D-R |
PFO, PSO, W-M |
medium |
[36] |
|
nanosized rice husk |
- |
76.92 |
- |
L, F, Te, D-R |
PFO, PSO, W-M |
medium |
[39] |
|
cotton plant ash |
2 |
0.64 |
- |
L, F |
PFO, PSO, W-M |
low |
[42] |
|
mustard plant ash |
- |
0.76 |
- |
L, F, Te |
PFO, PSO |
low |
[43] |
|
groundnut shell ash |
22 |
0.87 |
- |
L, F, Te |
PFO, PSO |
low |
[44] |
|
groundnut shell ash |
8 |
0.87 |
- |
L, F, Te |
PFO, PSO |
low |
[45] |
|
wheat straw ash |
37 |
1.89 |
- |
L, F, Te |
PFO, PSO |
low |
[46] |
|
functionalized Physalis peruviana biomass |
- |
233.3 |
- |
ITM |
- |
high |
[47] |
|
acid-treated peanut skin |
- |
246.7 |
- |
L, F, Te, D-R |
PFO, PSO, E |
high |
[48] |
|
apple shell |
- |
40.08 |
- |
L, F |
contact time |
medium |
[40] |
|
orange peel |
- |
22.71 |
- |
L, F |
contact time |
medium |
[40] |
|
banana peel |
- |
33.26 |
- |
L, F |
contact time |
medium |
[40] |
|
millet waste |
- |
45.45 |
- |
L, F |
contact time |
medium |
[40] |
|
coffee wastes |
- |
- |
340.0 |
L, F |
PFO, PSO |
high |
[41] |
|
N-cetylpyridinium-modified tiger nut residue |
0.03 |
79.3 |
- |
L, F, K-C |
PFO, PSO, E, W-M |
medium |
[50] |
|
H2SO4-modified wheat husks |
- |
161.1 |
- |
L, F |
Ba |
high |
[49] |
|
saw-sedge |
0.6 |
65.58 |
- |
L, F |
PFO, PSO |
medium |
[51] |
|
biomass-based MOF composite |
71 |
79.2 |
- |
L, F, Te, K-C |
PFO, PSO |
medium |
[52] |
|
biomass-based ferrospinel composite |
528 |
400 |
- |
L, F |
PFO, PSO, W-M |
high |
[53] |
“At the same time, it should be added that the presented review on the use of adsorbents based on waste materials for water purification, also showed the need to intensify research towards extending it to more complex systems corresponding to real ones. Only a few studies concern this problem, while, water and sewage contain many toxic substances with very diverse properties, originating from various sources, including those used as pesticide additives. The aim of adsorption techniques is to remove them to the highest possible extent, which is difficult to achieve in such complex systems in which a large part of the components may compete for access to adsorption centres. Therefore, an important direction of research is the analysis of adsorption in model and real multi-component systems. Taking into account the diversity of physicochemical properties of toxic substances present in water and sewage, including molecular size, hydrophobic or hydrophilic nature, and molecular form, it would also be necessary to analyze materials or systems of materials with different porosity and chemical nature of the surface in order to achieve the maximum possible level of purification. On the other hand in many cases the selective adsorbents should be applied, thus, also the studies concerning the mutual interactions between the adsorbates in complex systems which can decrease the adsorption effectiveness of a given substance are of great importance.”
(7. Conclusions)

Reviewer 2 Report
Comments and Suggestions for Authors
This paper provides a comprehensive overview of various adsorbents based on agricultural and household waste for chlo- 11 rophenoxy herbicides removal. It provides a fairly comprehensive field summary, but the introduction could be more specific in some sections.
C1- Lines 69-78, page 2 ; required references for each technique cited.
C2- The authors could consider adding the following articles as references, which would again increase the interest in general on applying biowaste for micropollutant adsorption :
https://doi.org/10.1016/j.molliq.2021.117092
https://doi.org/10.1080/01496395.2016.1231692
C3- The authors should add another section detailing the adsorption mechanisms proved in the literature for the bio-adsorption of Phenoxyacetic Herbicides.
C4- More details about the parameters to be considered for cost calculation could be provided in the last section (5. Cost of production of waste-based activated carbons )
C5- After the regeneration and reusability part Line 929, page 23, it would be nice if you could consider that the adsorption within columns or a continuous flow system allows the regeneration and reusability better as a process demanding for the industrial level.
C6- In the conclusion section, it is recommended to present the conclusions and prospects separately as bullet points to enhance clarity.
Comments on the Quality of English Language
Some grammatical style should be considered e,g
Lines 16-17, page1: it should be like "based on the adsorption capacity analysis "
Author Response
Manuscript Number: Sustainability-2718239
Title: Adsorbents Based on Agricultural and Household Waste for the Removal of Phenoxyacetic Herbicides from Water: a Comprehensive Review
Reviewer #2: Comments on: Sustainability-2718239
This paper provides a comprehensive overview of various adsorbents based on agricultural and household waste for chlorophenoxy herbicides removal. It provides a fairly comprehensive field summary, but the introduction could be more specific in some sections.
C1- Lines 69-78, page 2 ; required references for each technique cited.
Response to Reviewer 2 suggestion:
According to Reviewer’s suggestion the papers for each technique have been cited in the text.
“The most used water treatment techniques comprise adsorption [20], membrane filtration [21], irradiation [22], photocatalytic degradation [23], thermal remediation [24], electrokinetic coagulation [25], oxidation [26], ozonation [27], chemical coagulation-flocculation [28], ion exchange [29], and biodegradation [30].”
- Ighalo, J.O.; Ojukwu, V.E.; Umeh, C.T.; Aniagor, C.O.; Chinyelu, C.E.; Ajala, O.J.; Dulta, K.; Adeola, A.O.; Rangabhashiyam, S. Recent advances in the adsorptive removal of 2,4-dichlorophenoxyacetic acid from water. Journal of Water Process Engineering 2023, 56, 104514, doi:https://doi.org/10.1016/j.jwpe.2023.104514.
- Zhang, J.; Weston, G.; Yang, X.; Gray, S.; Duke, M. Removal of herbicide 2-methyl-4-chlorophenoxyacetic acid (MCPA) from saline industrial wastewater by reverse osmosis and nanofiltration. Desalination 2020, 496, 114691, doi:https://doi.org/10.1016/j.desal.2020.114691.
- Muszyński, P.; Brodowska, M.S.; Paszko, T. Occurrence and transformation of phenoxy acids in aquatic environment and photochemical methods of their removal: a review. Environmental Science and Pollution Research 2020, 27, 1276-1293, doi:10.1007/s11356-019-06510-2.
- Aziz, F.; Ouazzani, N.; Mandi, L.; Muhammad, M.; Uheida, A. Composite nanofibers of polyacrylonitrile/natural clay for decontamination of water containing Pb(II), Cu(II), Zn(II) and pesticides. Separation Science and Technology 2017, 52, 58-70, doi:10.1080/01496395.2016.1231692.
- Phorostianenko, O.; Petruk, H. THERMAL TREATMENT AS A METHOD OF REMEDIATION OF SOIL CONTAMINATED WITH PESTICIDES. Personality and Environmental Issues 2023, 3, 38-42, doi:10.31652/2786-6033-2023-2(4)-38-42.
- Brillas, E.; Boye, B.; Baños, M.; Calpe, J.; Garrido Ponce, J.A. Electrochemical degradation of chlorophenoxy and chlorobenzoic herbicides in acidic aqueous medium by the peroxi-coagulation method. Chemosphere 2003, 51, 227-235, doi:10.1016/S0045-6535(02)00836-6.
- Girón-Navarro, R.; Linares-Hernández, I.; Teutli-Sequeira, E.A.; Martínez-Miranda, V.; Santoyo-Tepole, F. Evaluation and comparison of advanced oxidation processes for the degradation of 2,4-dichlorophenoxyacetic acid (2,4-D): a review. Environmental Science and Pollution Research 2021, 28, 26325-26358, doi:10.1007/s11356-021-13730-y.
- Bradu, C.; Magureanu, M.; Parvulescu, V.I. Degradation of the chlorophenoxyacetic herbicide 2,4-D by plasma-ozonation system. Journal of Hazardous Materials 2017, 336, doi:10.1016/j.jhazmat.2017.04.050.
- Badawi, A.K.; Salama, R.S.; Mostafa, M.M.M. Natural-based coagulants/flocculants as sustainable market-valued products for industrial wastewater treatment: a review of recent developments. RSC Advances 2023, 13, 19335-19355, doi:10.1039/D3RA01999C.
- Mourid, E.H.; Lakraimi, M.; Legrouri, A. Preparation of well-structured hybrid material through ion exchange of chloride by 2,4,5-trichlorophenoxyacetic herbicide in a layered double hydroxide. Materials Chemistry and Physics 2022, 278, 125570, doi:https://doi.org/10.1016/j.matchemphys.2021.125570.
- Magnoli, K.; Carranza, C.; Aluffi, M.; Magnoli, C.; Barberis, C. Fungal biodegradation of chlorinated herbicides: an overview with an emphasis on 2,4-D in Argentina. Biodegradation 2023, 34, 199-214, doi:10.1007/s10532-023-10022-9.
C2- The authors could consider adding the following articles as references, which would again increase the interest in general on applying biowaste for micropollutant adsorption :
https://doi.org/10.1016/j.molliq.2021.117092
https://doi.org/10.1080/01496395.2016.1231692
Response to Reviewer 2 suggestion:
We would like to thank the Reviewer for pointing out interesting articles that may increase the scientific value of our paper. The articles have been cited in the following parts of the manuscript:
“The most used water treatment techniques comprise adsorption [20], membrane filtration [21], irradiation [22], photocatalytic degradation [23], thermal remediation [24], electrokinetic coagulation [25], oxidation [26], ozonation [27], chemical coagulation-flocculation [28], ion exchange [29], and biodegradation [30].”
- Aziz, F.; Ouazzani, N.; Mandi, L.; Muhammad, M.; Uheida, A. Composite nanofibers of polyacrylonitrile/natural clay for decontamination of water containing Pb(II), Cu(II), Zn(II) and pesticides. Separation Science and Technology 2017, 52, 58-70, doi:10.1080/01496395.2016.1231692.
“There are various groups of adsorbents i.e. (i) carbonaceous materials (commercial activated carbons, activated carbons from solid waste, carbon blacks, carbon nanotubes, graphene, reduced graphene oxide, and ordered mesoporous carbons); (ii) inorganic materials (silica gels, activated alumina, zeolites, molecular sieves, and metal oxides); (iii) advanced materials (inorganic-organic composites, doped materials); (iv) low-cost materials (agricultural and household waste-based materials, natural materials, biosorbents, industrial by-products) [33,34].”
- Aziz, K.; Aziz, F.; Mamouni, R.; Aziz, L.; Saffaj, N. Engineering of highly Brachychiton populneus shells@polyaniline bio-sorbent for efficient removal of pesticides from wastewater: Optimization using BBD-RSM approach. Journal of Molecular Liquids 2022, 346, 117092, doi:https://doi.org/10.1016/j.molliq.2021.117092.
C3- The authors should add another section detailing the adsorption mechanisms proved in the literature for the bio-adsorption of Phenoxyacetic Herbicides.
Response to Reviewer 2 suggestion:
The manuscript has been enriched by adding the following section:
“3. Mechanism of adsorption
Adsorption of chlorophenoxy herbicides on the different adsorbents based on agricultural and household waste is the result of various specific and nonspecific interactions. The adsorption mechanism of the same compound may change depending on the different properties of the adsorbent. The same situation occurs, for example, with a change in pH because the pH of the solution affects both the degree of dissociation and ionization of the adsorbate molecules, as well as the charge that will be present on the surface of the adsorbent. Thus, depending on the pH of the solution, adsorption can occur via electrostatic interactions (attractive or repulsive) between the adsorbate and the adsorbent.
The most typical and important forces driving adsorption on biochars and activated carbons include π-π interactions, electrostatic attraction and repulsion, hydrophobic interactions, covalent bonding and hydrogen bonding interactions between herbicide molecules and adsorbent surface functional groups. A significant share of micro- or small mesopores in the structure of these materials causes that adsorption proceeds mainly by the pore-filling mechanism. Figure 1a presents possible supramolecular systems created during adsorption of 2,4-D molecule on the exemplary biochar [66]. Impregnation of raw material with H3PO4 at the pre-treatment stage of synthesis determined the occurrence of typical interactions between the adsorbate and the adsorbent, as well as specific ones resulted from activating agent used (i.e. with metaphosphate groups). The great value of maximum adsorption capacity for this biochar towards 2,4-D (323.76 mg g−1) at pH = 2 and T = 318 K was found. Additionally, the morphological structure of used adsorbent and the herbicide adsorption isotherms are illustrated in Figures 1b-c and 1d, respectively.
Figure 1. The possible interactions in the 2,4-D adsorptive process using coffee ground char (a). SEM images of the adsorbent (b-c). The herbicide adsorption isotherms on biochar (d). [66].
Regarding unmodified, modified natural materials and ashes, mechanism of the 2,4-D or MCPA adsorption must be considered individually in conjunction with unique (individual) physicochemical properties of the adsorbent. According to the authors of some works [35-37,39] adsorption of chlorophenoxy herbicides on rice husk with positive surface charge, follows a chemical ion-exchange mechanism or via a chemical process (involving valency force sharing). Chemical reactions (through free electron pairs of surface functional groups e.g., carbonyl, amide, ether), cation–dipole interactions (due to the occurrence of hydroxyl, amine, and amide groups) as well as hydrogen bonds are proposed for adsorption of 2,4-D on the saw-sedge (C. mariscus). In turn, for bio-waste materials such as apple shell, orange peel, banana peel, and millet residue, the discussed process results of molecular interactions (when the molecular form of the pesticide occurs) and repulsive electrostatic interactions (at solution pH where both the adsorbent surface moieties i.e. carboxyl and sulphate, and 2,4-D are negatively charged) [40]. Al-Zaben and Mekhamer [41] reported no relationship between the MCPA removal efficiency and the pH solution for coffee waste as adsorbent. The existence of electrostatic interaction was excluded in this system, and the responsibility for adsorption was assigned to hydrogen bonds between OH moieties on the adsorbent surface and acidic group of adsorbate molecules.
According to authors of papers [47-49] oxidation of various biomass (Physalis peruviana fruit, Arachis hypogaea skins and wheat husks) with sulfuric acid resulted in oxygenated functionalities on their surface which participated in adsorption of the chlorophenoxy herbicides via electrostatic forces, hydrogen bonding and van der Waals forces. The authors of the latter paper [49] indicated also halogen bonds and π–π interactions as responsible for adsorption. Noteworthy is the relatively high adsorption efficiency of acid-modified wheat husks, with the capacity of 161.1 mg g−1 (pH = 2, T = 298 K). For this reason, the possible supramolecular systems of this adsorbent with the 2,4-D molecule (Figure 2a) along with the morphological structure of adsorbent (Figure 2b-d), and the equilibrium isotherms at various experimental temperatures (Figure 2e) are presented.
Figure 2. The possible interactions in the 2,4-D adsorptive process using acid-modified wheat husks (a). SEM images of adsorbent (magnitudes (b) × 500, (c) × 1000 (d) x 3000) (b-d). Equilibrium isotherms of the 2,4-D adsorption on modified wheat husks (pH = 2) (e) [49].
Modification of tiger nut residue using N-cetylpyridinium [50] resulted in obtaining the adsorbent, which interacts with 2,4-D through electrostatic attraction, hydrogen bonding and a triggered π-π stacking. Trivedi's comprehensive research [42-46] on ashes originated from various waste materials showed a close relationship between their chemical composition and the mechanism of the chlorophenoxy herbicides adsorption. The presence of metallic oxides (CaO, K2O) in ash, which in water take a form of metallic hydroxides resulted in electrostatic repulsion forces between them and adsorbate anions. In turn, for Al2O3 which constitutes positive centers on the adsorbent surface, the trigger attractive electrostatic interactions were observed. Thus, the mineral composition of ash determines mechanism and effectiveness of the adsorption process. For hydrochars (HCs) mechanism of the 2,4-D adsorption via intensive partitioning and/or chemisorption was postulated [64]. Regarding the interactions of pollutants with more advanced magnetic waste-based materials [52,53] with various functional groups (hydroxyl, amine, carboxyl), the hydrogen bonding, electrostatic complexation, and π-π mechanisms were suggested. Moreover, some of these interactions were additionally enhanced by the occurrence of iron. The high efficiency of ferrospinel composite (FLPWAC) [53] (qm = 400 mg g−1, T=318 K) was found. The possible supramolecular adsorbate-adsorbent systems (Figure 3a), the morphological structure of adsorbent (Figure 3b-f), and the equilibrium isotherms at various experimental temperatures (Figure 3g) are presented.
Figure 3. The possible interactions in the 2,4-D adsorptive process using the ferrospinel composite (a). SEM images of adsorbent at different expansion ratios (b-f). Langmuir plots of the 2,4-D adsorption on composite (g) [53].
Summing up, the variety of types of adsorbents and their different characteristics translate into a multitude of mechanisms leading to the adsorption of chlorophenoxy herbicides.”
C4- More details about the parameters to be considered for cost calculation could be provided in the last section (5. Cost of production of waste-based activated carbons)
Response to Reviewer 2 suggestion:
The text on production cost of waste-based activated carbons has been enriched with more detailed parameters:
“In a paper from 2022 [97] the authors estimated the production cost of Capsicum pulp-based AC at ~$ 1.75 per kg (summing up the costs of raw material transport; reagents - nitrogen gas, KOH, hydrochloric acid; and media - power, water) which is lower than that for commercial ACs, e.g. $ 2–5 per kg. In other works from 2021, fabrication of ACs from mushroom residues [89] and date palm coir waste [98] were assessed to be $ 2.39 per kg (the total costs of reagents - nitrogen gas, hydrochloric acid, zinc chloride; and media – power) and $ 3 per kg (the total costs of the capital expenses –, installation equipment, land, buildings, service; and operating ones (precursor collection, water, electricity, labor), respectively.”
C5- After the regeneration and reusability part Line 929, page 23, it would be nice if you could consider that the adsorption within columns or a continuous flow system allows the regeneration and reusability better as a process demanding for the industrial level.
Response to Reviewer 2 suggestion:
The text in the section of Regeneration and reusability of waste-based activated carbons was changed following the reviewer suggestion:
“All these findings showed that ACs successfully passed the regeneration and reusability tests. However, for application of adsorbents on an industrial scale, it is advisable to perform additional adsorption studies within column or a continuous flow system (fluidized bed column) for treatment of water polluted with chlorophenoxy herbicides.”
C6- In the conclusion section, it is recommended to present the conclusions and prospects separately as bullet points to enhance clarity.
Response to Reviewer 2 suggestion:
The Conclusions section was widened according to Reviewer suggestion by presenting the perspective trends in a separate part.
“At the same time, it should be added that the presented review on the use of adsorbents based on waste materials for water purification, also showed the need to intensify research towards extending it to more complex systems corresponding to real ones. Only a few studies concern this problem, while, water and sewage contain many toxic substances with very diverse properties, originating from various sources, including those used as pesticide additives. The aim of adsorption techniques is to remove them to the highest possible extent, which is difficult to achieve in such complex systems in which a large part of the components may compete for access to adsorption centres. Therefore, an important direction of research is the analysis of adsorption in model and real multi-component systems. Taking into account the diversity of physicochemical properties of toxic substances present in water and sewage, including molecular size, hydrophobic or hydrophilic nature, and molecular form, it would also be necessary to analyze materials or systems of materials with different porosity and chemical nature of the surface in order to achieve the maximum possible level of purification. On the other hand in many cases the selective adsorbents should be applied, thus, also the studies concerning the mutual interactions between the adsorbates in complex systems which can decrease the adsorption effectiveness of a given substance are of great importance.”
Comments on the Quality of English Language
Some grammatical style should be considered e,g
Lines 16-17, page1: it should be like "based on the adsorption capacity analysis "
Response to Reviewer 2 suggestion:
The manuscript was checked and corrected.

Reviewer 3 Report
Comments and Suggestions for Authors
I think the manuscript may be accepted for publication in Sustainability after the following minor revisions:
1) In Tables 2 to 4, besides adsorption capacity, another important adsorptivity or removal efficiency of phenoxyacetic herbicides as well as cost-benefit of the adsorbents should be added into those three Tables. In order to do this successfully, please refer to and cite the following references:
[1] Highly cost-efficient sorption and desorption of mercury ions onto regenerable poly(m-phenylenediamine) microspheres with many active groups. Chemical Engineering Journal 391 (2020) 123515.
[2] Synthesis of poly(1,5-diaminonaphthalene) microparticles with abundant amino and imino groups as strong adsorbers for heavy metal ions. Microchim Acta 186 (2019) 208.
2) The molecular and supramolecular structures of various adsorbents might be discussed or elaborated to find the truly high-performance adsorbents towards phenoxyacetic herbicides.
3) The morphological structures of representative advanced adsorbents of phenoxyacetic herbicides should also be shown as images, if possible.
Comments on the Quality of English LanguageThe article title seems slightly lengthy and could be polished or shortened to some extent.
Author Response
Manuscript Number: Sustainability-2718239
Title: Adsorbents Based on Agricultural and Household Waste for the Removal of Phenoxyacetic Herbicides from Water: a Comprehensive Review
Reviewer #3: Comments on: Sustainability-2718239
I think the manuscript may be accepted for publication in Sustainability after the following minor revisions:
1) In Tables 2 to 4, besides adsorption capacity, another important adsorptivity or removal efficiency of phenoxyacetic herbicides as well as cost-benefit of the adsorbents should be added into those three Tables. In order to do this successfully, please refer to and cite the following references:
[1] Highly cost-efficient sorption and desorption of mercury ions onto regenerable poly(m-phenylenediamine) microspheres with many active groups. Chemical Engineering Journal 391 (2020) 123515.
[2] Synthesis of poly(1,5-diaminonaphthalene) microparticles with abundant amino and imino groups as strong adsorbers for heavy metal ions. Microchim Acta 186 (2019) 208.
Response to Reviewer 3 suggestion:
Reviewer’s suggestion was considered and evaluation of cost-benefit of the adsorbents in tables 2-4 was added. Information about it was given in the following text:
“Various unmodified, modified agricultural and household waste materials and ashes, and their adsorption performance in chlorophenoxy herbicides removal are summarized in Table 2. Additionally, the cost-benefit of the given adsorbents was evaluated based on the comparison of their adsorption capacity towards chlorophenoxy herbicides with synthesis cost, the impact of their production on the natural environment (energy consumption, harmfulness of reagents used, secondary waste) and their recyclability. Such subjective evaluation is widely accepted and practiced by many scientists [54,55] “
- Li, X.-G.; Huang, M.-R.; Tao, T.; Ren, Z.; Zeng, J.; Yu, J.; Umeyama, T.; Ohara, T.; Imahori, H. Highly cost-efficient sorption and desorption of mercury ions onto regenerable poly(m-phenylenediamine) microspheres with many active groups. Chemical Engineering Journal 2020, 391, 123515, doi:https://doi.org/10.1016/j.cej.2019.123515.
- Li, X.-G.; Huang, M.-R.; Jiang, Y.-B.; Yu, J.; He, Z. Synthesis of poly(1,5-diaminonaphthalene) microparticles with abundant amino and imino groups as strong adsorbers for heavy metal ions. Microchimica Acta 2019, 186, 208, doi:10.1007/s00604-019-3330-z.
Due to different experimental parameters (i.e. adsorbent mass, initial solution concentration, solution volume, solution pH) used by researchers for determining the removal efficiency of phenoxyacetic herbicides, this value for various adsorbents cannot be compared. It requires that the adsorption studies maintain the same experimental parameters.
2) The molecular and supramolecular structures of various adsorbents might be discussed or elaborated to find the truly high-performance adsorbents towards phenoxyacetic herbicides.
3) The morphological structures of representative advanced adsorbents of phenoxyacetic herbicides should also be shown as images, if possible.
Response to Reviewer 3 suggestions 2) and 3):
The manuscript was enriched by adding the following section, which includes the molecular, supramolecular and morphological structures of the exemplary adsorbents along with a discussion:
“3. Mechanism of adsorption
Adsorption of chlorophenoxy herbicides on the different adsorbents based on agricultural and household waste is the result of various specific and nonspecific interactions. The adsorption mechanism of the same compound may change depending on the different properties of the adsorbent. The same situation occurs, for example, with a change in pH because the pH of the solution affects both the degree of dissociation and ionization of the adsorbate molecules, as well as the charge that will be present on the surface of the adsorbent. Thus, depending on the pH of the solution, adsorption can occur via electrostatic interactions (attractive or repulsive) between the adsorbate and the adsorbent.
The most typical and important forces driving adsorption on biochars and activated carbons include π-π interactions, electrostatic attraction and repulsion, hydrophobic interactions, covalent bonding and hydrogen bonding interactions between herbicide molecules and adsorbent surface functional groups. A significant share of micro- or small mesopores in the structure of these materials causes that adsorption proceeds mainly by the pore-filling mechanism. Figure 1a presents possible supramolecular systems created during adsorption of 2,4-D molecule on the exemplary biochar [66]. Impregnation of raw material with H3PO4 at the pre-treatment stage of synthesis determined the occurrence of typical interactions between the adsorbate and the adsorbent, as well as specific ones resulted from activating agent used (i.e. with metaphosphate groups). The great value of maximum adsorption capacity for this biochar towards 2,4-D (323.76 mg g−1) at pH = 2 and T = 318 K was found. Additionally, the morphological structure of used adsorbent and the herbicide adsorption isotherms are illustrated in Figures 1b-c and 1d, respectively.
Figure 1. The possible interactions in the 2,4-D adsorptive process using coffee ground char (a). SEM images of the adsorbent (b-c). The herbicide adsorption isotherms on biochar (d). [66].
Regarding unmodified, modified natural materials and ashes, mechanism of the 2,4-D or MCPA adsorption must be considered individually in conjunction with unique (individual) physicochemical properties of the adsorbent. According to the authors of some works [35-37,39] adsorption of chlorophenoxy herbicides on rice husk with positive surface charge, follows a chemical ion-exchange mechanism or via a chemical process (involving valency force sharing). Chemical reactions (through free electron pairs of surface functional groups e.g., carbonyl, amide, ether), cation–dipole interactions (due to the occurrence of hydroxyl, amine, and amide groups) as well as hydrogen bonds are proposed for adsorption of 2,4-D on the saw-sedge (C. mariscus). In turn, for bio-waste materials such as apple shell, orange peel, banana peel, and millet residue, the discussed process results of molecular interactions (when the molecular form of the pesticide occurs) and repulsive electrostatic interactions (at solution pH where both the adsorbent surface moieties i.e. carboxyl and sulphate, and 2,4-D are negatively charged) [40]. Al-Zaben and Mekhamer [41] reported no relationship between the MCPA removal efficiency and the pH solution for coffee waste as adsorbent. The existence of electrostatic interaction was excluded in this system, and the responsibility for adsorption was assigned to hydrogen bonds between OH moieties on the adsorbent surface and acidic group of adsorbate molecules.
According to authors of papers [47-49] oxidation of various biomass (Physalis peruviana fruit, Arachis hypogaea skins and wheat husks) with sulfuric acid resulted in oxygenated functionalities on their surface which participated in adsorption of the chlorophenoxy herbicides via electrostatic forces, hydrogen bonding and van der Waals forces. The authors of the latter paper [49] indicated also halogen bonds and π–π interactions as responsible for adsorption. Noteworthy is the relatively high adsorption efficiency of acid-modified wheat husks, with the capacity of 161.1 mg g−1 (pH = 2, T = 298 K). For this reason, the possible supramolecular systems of this adsorbent with the 2,4-D molecule (Figure 2a) along with the morphological structure of adsorbent (Figure 2b-d), and the equilibrium isotherms at various experimental temperatures (Figure 2e) are presented.
Figure 2. The possible interactions in the 2,4-D adsorptive process using acid-modified wheat husks (a). SEM images of adsorbent (magnitudes (b) × 500, (c) × 1000 (d) x 3000) (b-d). Equilibrium isotherms of the 2,4-D adsorption on modified wheat husks (pH = 2) (e) [49].
Modification of tiger nut residue using N-cetylpyridinium [50] resulted in obtaining the adsorbent, which interacts with 2,4-D through electrostatic attraction, hydrogen bonding and a triggered π-π stacking. Trivedi's comprehensive research [42-46] on ashes originated from various waste materials showed a close relationship between their chemical composition and the mechanism of the chlorophenoxy herbicides adsorption. The presence of metallic oxides (CaO, K2O) in ash, which in water take a form of metallic hydroxides resulted in electrostatic repulsion forces between them and adsorbate anions. In turn, for Al2O3 which constitutes positive centers on the adsorbent surface, the trigger attractive electrostatic interactions were observed. Thus, the mineral composition of ash determines mechanism and effectiveness of the adsorption process. For hydrochars (HCs) mechanism of the 2,4-D adsorption via intensive partitioning and/or chemisorption was postulated [64]. Regarding the interactions of pollutants with more advanced magnetic waste-based materials [52,53] with various functional groups (hydroxyl, amine, carboxyl), the hydrogen bonding, electrostatic complexation, and π-π mechanisms were suggested. Moreover, some of these interactions were additionally enhanced by the occurrence of iron. The high efficiency of ferrospinel composite (FLPWAC) [53] (qm = 400 mg g−1, T=318 K) was found. The possible supramolecular adsorbate-adsorbent systems (Figure 3a), the morphological structure of adsorbent (Figure 3b-f), and the equilibrium isotherms at various experimental temperatures (Figure 3g) are presented.
Figure 3. The possible interactions in the 2,4-D adsorptive process using the ferrospinel composite (a). SEM images of adsorbent at different expansion ratios (b-f). Langmuir plots of the 2,4-D adsorption on composite (g) [53].
Summing up, the variety of types of adsorbents and their different characteristics translate into a multitude of mechanisms leading to the adsorption of chlorophenoxy herbicides.”
Comments on the Quality of English Language
The article title seems slightly lengthy and could be polished or shortened to some extent.
Response to Reviewer 3 suggestion:
The title was changed following the Reviewer suggestion:
Title: Waste-Based Adsorbents for the Removal of Phenoxyacetic Herbicides from Water: a Comprehensive Review

Reviewer 4 Report
Comments and Suggestions for Authors
The paper must be completed with adsorption data in form of isotherms and
also study the interaction mechanism between the adsorbents and the adsorbant.
It has to be modified since the chemical point of view and add figures to show more clearly the results.
Author Response
Manuscript Number: Sustainability-2718239
Title: Adsorbents Based on Agricultural and Household Waste for the Removal of Phenoxyacetic Herbicides from Water: a Comprehensive Review
Reviewer #4: Comments on: Sustainability-2718239
The paper must be completed with adsorption data in form of isotherms and also study the interaction mechanism between the adsorbents and the adsorbate. It has to be modified since the chemical point of view and add figures to show more clearly the results.
Response to Reviewer 4 suggestion:
We would like to thank the Reviewer for pointing out the remarks that may increase the scientific value of our paper. According to Reviewer suggestion three selected adsorption isotherms have been presented as the representatives of a large number of adsorption systems described in the manuscript. Moreover, the discussion of adsorption mechanisms for various systems herbicide – waste based adsorbent has been added. Additionally, the supramolecular and morphological structures of the selected adsorbents have been also presented. The manuscript has been enriched by adding the following section:
“3. Mechanism of adsorption
Adsorption of chlorophenoxy herbicides on the different adsorbents based on agricultural and household waste is the result of various specific and nonspecific interactions. The adsorption mechanism of the same compound may change depending on the different properties of the adsorbent. The same situation occurs, for example, with a change in pH because the pH of the solution affects both the degree of dissociation and ionization of the adsorbate molecules, as well as the charge that will be present on the surface of the adsorbent. Thus, depending on the pH of the solution, adsorption can occur via electrostatic interactions (attractive or repulsive) between the adsorbate and the adsorbent.
The most typical and important forces driving adsorption on biochars and activated carbons include π-π interactions, electrostatic attraction and repulsion, hydrophobic interactions, covalent bonding and hydrogen bonding interactions between herbicide molecules and adsorbent surface functional groups. A significant share of micro- or small mesopores in the structure of these materials causes that adsorption proceeds mainly by the pore-filling mechanism. Figure 1a presents possible supramolecular systems created during adsorption of 2,4-D molecule on the exemplary biochar [66]. Impregnation of raw material with H3PO4 at the pre-treatment stage of synthesis determined the occurrence of typical interactions between the adsorbate and the adsorbent, as well as specific ones resulted from activating agent used (i.e. with metaphosphate groups). The great value of maximum adsorption capacity for this biochar towards 2,4-D (323.76 mg g−1) at pH = 2 and T = 318 K was found. Additionally, the morphological structure of used adsorbent and the herbicide adsorption isotherms are illustrated in Figures 1b-c and 1d, respectively.
Figure 1. The possible interactions in the 2,4-D adsorptive process using coffee ground char (a). SEM images of the adsorbent (b-c). The herbicide adsorption isotherms on biochar (d). [66].
Regarding unmodified, modified natural materials and ashes, mechanism of the 2,4-D or MCPA adsorption must be considered individually in conjunction with unique (individual) physicochemical properties of the adsorbent. According to the authors of some works [35-37,39] adsorption of chlorophenoxy herbicides on rice husk with positive surface charge, follows a chemical ion-exchange mechanism or via a chemical process (involving valency force sharing). Chemical reactions (through free electron pairs of surface functional groups e.g., carbonyl, amide, ether), cation–dipole interactions (due to the occurrence of hydroxyl, amine, and amide groups) as well as hydrogen bonds are proposed for adsorption of 2,4-D on the saw-sedge (C. mariscus). In turn, for bio-waste materials such as apple shell, orange peel, banana peel, and millet residue, the discussed process results of molecular interactions (when the molecular form of the pesticide occurs) and repulsive electrostatic interactions (at solution pH where both the adsorbent surface moieties i.e. carboxyl and sulphate, and 2,4-D are negatively charged) [40]. Al-Zaben and Mekhamer [41] reported no relationship between the MCPA removal efficiency and the pH solution for coffee waste as adsorbent. The existence of electrostatic interaction was excluded in this system, and the responsibility for adsorption was assigned to hydrogen bonds between OH moieties on the adsorbent surface and acidic group of adsorbate molecules.
According to authors of papers [47-49] oxidation of various biomass (Physalis peruviana fruit, Arachis hypogaea skins and wheat husks) with sulfuric acid resulted in oxygenated functionalities on their surface which participated in adsorption of the chlorophenoxy herbicides via electrostatic forces, hydrogen bonding and van der Waals forces. The authors of the latter paper [49] indicated also halogen bonds and π–π interactions as responsible for adsorption. Noteworthy is the relatively high adsorption efficiency of acid-modified wheat husks, with the capacity of 161.1 mg g−1 (pH = 2, T = 298 K). For this reason, the possible supramolecular systems of this adsorbent with the 2,4-D molecule (Figure 2a) along with the morphological structure of adsorbent (Figure 2b-d), and the equilibrium isotherms at various experimental temperatures (Figure 2e) are presented.
Figure 2. The possible interactions in the 2,4-D adsorptive process using acid-modified wheat husks (a). SEM images of adsorbent (magnitudes (b) × 500, (c) × 1000 (d) x 3000) (b-d). Equilibrium isotherms of the 2,4-D adsorption on modified wheat husks (pH = 2) (e) [49].
Modification of tiger nut residue using N-cetylpyridinium [50] resulted in obtaining the adsorbent, which interacts with 2,4-D through electrostatic attraction, hydrogen bonding and a triggered π-π stacking. Trivedi's comprehensive research [42-46] on ashes originated from various waste materials showed a close relationship between their chemical composition and the mechanism of the chlorophenoxy herbicides adsorption. The presence of metallic oxides (CaO, K2O) in ash, which in water take a form of metallic hydroxides resulted in electrostatic repulsion forces between them and adsorbate anions. In turn, for Al2O3 which constitutes positive centers on the adsorbent surface, the trigger attractive electrostatic interactions were observed. Thus, the mineral composition of ash determines mechanism and effectiveness of the adsorption process. For hydrochars (HCs) mechanism of the 2,4-D adsorption via intensive partitioning and/or chemisorption was postulated [64]. Regarding the interactions of pollutants with more advanced magnetic waste-based materials [52,53] with various functional groups (hydroxyl, amine, carboxyl), the hydrogen bonding, electrostatic complexation, and π-π mechanisms were suggested. Moreover, some of these interactions were additionally enhanced by the occurrence of iron. The high efficiency of ferrospinel composite (FLPWAC) [53] (qm = 400 mg g−1, T=318 K) was found. The possible supramolecular adsorbate-adsorbent systems (Figure 3a), the morphological structure of adsorbent (Figure 3b-f), and the equilibrium isotherms at various experimental temperatures (Figure 3g) are presented.
Figure 3. The possible interactions in the 2,4-D adsorptive process using the ferrospinel composite (a). SEM images of adsorbent at different expansion ratios (b-f). Langmuir plots of the 2,4-D adsorption on composite (g) [53].
Summing up, the variety of types of adsorbents and their different characteristics translate into a multitude of mechanisms leading to the adsorption of chlorophenoxy herbicides.”

Reviewer 5 Report
Comments and Suggestions for Authors
I have now completed reviewing the paper entitled "Adsorbents Based on Agricultural and Household Waste for the Removal of Phenoxyacetic Herbicides from Water: A Comprehensive Review." Authors don't go any further in their analysis to add more knowledge to the subject than what is already known from scientific literature. Authors should consider analyzing these findings from the first principle with the support of applicable equations. The paper was supposed to be a one-stop contact for adsorbents based on agricultural and household waste for the removal of phenolacetic herbicides from water. However, it is not clear in terms of the efficiency of processes employing adsorbents based on agricultural and household waste, their recyclability, and so forth.
Comments on the Quality of English LanguageThere are a lot of grammar errors that jeopardize the quality of the work done, and major revisions should be considered prior to publication.
Author Response
Manuscript Number: Sustainability-2718239
Title: Adsorbents Based on Agricultural and Household Waste for the Removal of Phenoxyacetic Herbicides from Water: a Comprehensive Review
Reviewer #5: Comments on: Sustainability-2718239
I have now completed reviewing the paper entitled "Adsorbents Based on Agricultural and Household Waste for the Removal of Phenoxyacetic Herbicides from Water: A Comprehensive Review." Authors don't go any further in their analysis to add more knowledge to the subject than what is already known from scientific literature. Authors should consider analyzing these findings from the first principle with the support of applicable equations. The paper was supposed to be a one-stop contact for adsorbents based on agricultural and household waste for the removal of phenolacetic herbicides from water.
Response to Reviewer 5 suggestions:
The article presents a wide range of agricultural and household wastes as low-cost alternative adsorbents for the removal of two commonly used herbicides (2,4-D and MCPA) from water. The characteristics of the adsorbents, the conditions for adsorption experiments and their results, and the theoretical models that best describe the adsorption process have been described. Based on recent literature, the suitability and effectiveness of waste materials as low-cost adsorbents, both in their original form and after modification, mainly to waste biocarbon and activated carbon, have been demonstrated. The following chapters discuss the effect of co-substances on the adsorption of chlorophenoxy herbicides on waste-based adsorbents, as well as the problems of regeneration and reusability of waste-based activated carbons and the associated costs. The waste materials presented in this work have shown a satisfactory affinity for 2,4-D and MCPA, and since they are relatively cheap (or completely free), readily available, and easily modified, e.g. by pyrolysis and activation, their use as alternative adsorbents seems very interesting and promising. The use of the waste itself as an adsorbent reduces the cost of waste disposal and thus promotes environmental protection, and the low cost of these materials means that they can be disposed of after use without costly regeneration.
In our work, we have clearly demonstrated that the development of adsorbents using agricultural and household wastes is an interesting alternative to commercial materials for environmentally friendly applications, and at the same time a way to manage them properly. We indicated the potential areas for further research and emerging trends.
As for a novelty of this article, a large number of publications on the use of waste materials for adsorption, especially in the context of the removal of chlorophenoxy herbicides, indicates the topicality of the subject and the great interest of many research centers (and therefore the accessibility to a wide range of readers).
Such a large number of publications (and thus a large number of different adsorbents used) means that this knowledge is indeed, as the reviewer claims, well established and extensively researched, but unfortunately very scattered. In our review, we have set ourselves the task of bringing together the current state of knowledge on this subject in one place. In this respect, we believe that this is a very valuable and attractive project for potential readers. The presentation of the current state of knowledge and the comparison of many different adsorbents may help many researchers to take their research forward. Despite the great interest in the topic of adsorption of phenoxyacetic herbicides from water and a large number of published research articles, there are no valuable review papers that organize the current developments and the current state of knowledge. To the best of our knowledge, the reviewed paper would be only the second of its kind. The first review (and the only one to our knowledge) concerns the adsorption of phenoxyacetic herbicides from water on carbonaceous and non-carbonaceous adsorbents (Molecules 28 (2023) 5404. https://doi.org/10.3390/molecules28145404).
Regarding the Reviewer suggestions the manuscript was enriched by adding the following section:
“3. Mechanism of adsorption
Adsorption of chlorophenoxy herbicides on the different adsorbents based on agricultural and household waste is the result of various specific and nonspecific interactions. The adsorption mechanism of the same compound may change depending on the different properties of the adsorbent. The same situation occurs, for example, with a change in pH because the pH of the solution affects both the degree of dissociation and ionization of the adsorbate molecules, as well as the charge that will be present on the surface of the adsorbent. Thus, depending on the pH of the solution, adsorption can occur via electrostatic interactions (attractive or repulsive) between the adsorbate and the adsorbent.
The most typical and important forces driving adsorption on biochars and activated carbons include π-π interactions, electrostatic attraction and repulsion, hydrophobic interactions, covalent bonding and hydrogen bonding interactions between herbicide molecules and adsorbent surface functional groups. A significant share of micro- or small mesopores in the structure of these materials causes that adsorption proceeds mainly by the pore-filling mechanism. Figure 1a presents possible supramolecular systems created during adsorption of 2,4-D molecule on the exemplary biochar [66]. Impregnation of raw material with H3PO4 at the pre-treatment stage of synthesis determined the occurrence of typical interactions between the adsorbate and the adsorbent, as well as specific ones resulted from activating agent used (i.e. with metaphosphate groups). The great value of maximum adsorption capacity for this biochar towards 2,4-D (323.76 mg g−1) at pH = 2 and T = 318 K was found. Additionally, the morphological structure of used adsorbent and the herbicide adsorption isotherms are illustrated in Figures 1b-c and 1d, respectively.
Figure 1. The possible interactions in the 2,4-D adsorptive process using coffee ground char (a). SEM images of the adsorbent (b-c). The herbicide adsorption isotherms on biochar (d). [66].
Regarding unmodified, modified natural materials and ashes, mechanism of the 2,4-D or MCPA adsorption must be considered individually in conjunction with unique (individual) physicochemical properties of the adsorbent. According to the authors of some works [35-37,39] adsorption of chlorophenoxy herbicides on rice husk with positive surface charge, follows a chemical ion-exchange mechanism or via a chemical process (involving valency force sharing). Chemical reactions (through free electron pairs of surface functional groups e.g., carbonyl, amide, ether), cation–dipole interactions (due to the occurrence of hydroxyl, amine, and amide groups) as well as hydrogen bonds are proposed for adsorption of 2,4-D on the saw-sedge (C. mariscus). In turn, for bio-waste materials such as apple shell, orange peel, banana peel, and millet residue, the discussed process results of molecular interactions (when the molecular form of the pesticide occurs) and repulsive electrostatic interactions (at solution pH where both the adsorbent surface moieties i.e. carboxyl and sulphate, and 2,4-D are negatively charged) [40]. Al-Zaben and Mekhamer [41] reported no relationship between the MCPA removal efficiency and the pH solution for coffee waste as adsorbent. The existence of electrostatic interaction was excluded in this system, and the responsibility for adsorption was assigned to hydrogen bonds between OH moieties on the adsorbent surface and acidic group of adsorbate molecules.
According to authors of papers [47-49] oxidation of various biomass (Physalis peruviana fruit, Arachis hypogaea skins and wheat husks) with sulfuric acid resulted in oxygenated functionalities on their surface which participated in adsorption of the chlorophenoxy herbicides via electrostatic forces, hydrogen bonding and van der Waals forces. The authors of the latter paper [49] indicated also halogen bonds and π–π interactions as responsible for adsorption. Noteworthy is the relatively high adsorption efficiency of acid-modified wheat husks, with the capacity of 161.1 mg g−1 (pH = 2, T = 298 K). For this reason, the possible supramolecular systems of this adsorbent with the 2,4-D molecule (Figure 2a) along with the morphological structure of adsorbent (Figure 2b-d), and the equilibrium isotherms at various experimental temperatures (Figure 2e) are presented.
Figure 2. The possible interactions in the 2,4-D adsorptive process using acid-modified wheat husks (a). SEM images of adsorbent (magnitudes (b) × 500, (c) × 1000 (d) x 3000) (b-d). Equilibrium isotherms of the 2,4-D adsorption on modified wheat husks (pH = 2) (e) [49].
Modification of tiger nut residue using N-cetylpyridinium [50] resulted in obtaining the adsorbent, which interacts with 2,4-D through electrostatic attraction, hydrogen bonding and a triggered π-π stacking. Trivedi's comprehensive research [42-46] on ashes originated from various waste materials showed a close relationship between their chemical composition and the mechanism of the chlorophenoxy herbicides adsorption. The presence of metallic oxides (CaO, K2O) in ash, which in water take a form of metallic hydroxides resulted in electrostatic repulsion forces between them and adsorbate anions. In turn, for Al2O3 which constitutes positive centers on the adsorbent surface, the trigger attractive electrostatic interactions were observed. Thus, the mineral composition of ash determines mechanism and effectiveness of the adsorption process. For hydrochars (HCs) mechanism of the 2,4-D adsorption via intensive partitioning and/or chemisorption was postulated [64]. Regarding the interactions of pollutants with more advanced magnetic waste-based materials [52,53] with various functional groups (hydroxyl, amine, carboxyl), the hydrogen bonding, electrostatic complexation, and π-π mechanisms were suggested. Moreover, some of these interactions were additionally enhanced by the occurrence of iron. The high efficiency of ferrospinel composite (FLPWAC) [53] (qm = 400 mg g−1, T=318 K) was found. The possible supramolecular adsorbate-adsorbent systems (Figure 3a), the morphological structure of adsorbent (Figure 3b-f), and the equilibrium isotherms at various experimental temperatures (Figure 3g) are presented.
Figure 3. The possible interactions in the 2,4-D adsorptive process using the ferrospinel composite (a). SEM images of adsorbent at different expansion ratios (b-f). Langmuir plots of the 2,4-D adsorption on composite (g) [53].
Summing up, the variety of types of adsorbents and their different characteristics translate into a multitude of mechanisms leading to the adsorption of chlorophenoxy herbicides.”
The Conclusions section was widened by presenting the perspective trends.
“At the same time, it should be added that the presented review on the use of adsorbents based on waste materials for water purification, also showed the need to intensify research towards extending it to more complex systems corresponding to real ones. Only a few studies concern this problem, while, water and sewage contain many toxic substances with very diverse properties, originating from various sources, including those used as pesticide additives. The aim of adsorption techniques is to remove them to the highest possible extent, which is difficult to achieve in such complex systems in which a large part of the components may compete for access to adsorption centres. Therefore, an important direction of research is the analysis of adsorption in model and real multi-component systems. Taking into account the diversity of physicochemical properties of toxic substances present in water and sewage, including molecular size, hydrophobic or hydrophilic nature, and molecular form, it would also be necessary to analyze materials or systems of materials with different porosity and chemical nature of the surface in order to achieve the maximum possible level of purification. On the other hand in many cases the selective adsorbents should be applied, thus, also the studies concerning the mutual interactions between the adsorbates in complex systems which can decrease the adsorption effectiveness of a given substance are of great importance.”
However, it is not clear in terms of the efficiency of processes employing adsorbents based on agricultural and household waste, their recyclability, and so forth.
Response to Reviewer 5 suggestion:
Reviewer’s suggestion was considered and evaluation of cost-benefit of the adsorbents in tables 2-4 was added. Information about it was given in the following text:
“Various unmodified, modified agricultural and household waste materials and ashes, and their adsorption performance in chlorophenoxy herbicides removal are summarized in Table 2. Additionally, the cost-benefit of the given adsorbents was evaluated based on the comparison of their adsorption capacity towards chlorophenoxy herbicides with synthesis cost, the impact of their production on the natural environment (energy consumption, harmfulness of reagents used, secondary waste) and their recyclability. Such subjective evaluation is widely accepted and practiced by many scientists [54,55] “
The exemplary Table:
|
Adsorbent |
SBET m2 g−1 |
Adsorption capacity (qm) mg g−1 |
Isotherm model |
Kinetic model |
Integral cost - benefit |
Ref. |
|
|
2,4-D |
MCPA |
||||||
|
rice husk ash |
34 |
1.425 |
- |
L, F, Te |
PFO, PSO, E |
low |
[35] |
|
rice husk ash |
34 |
- |
1.681 |
L, F, Te |
PFO, PSO, E |
low |
[37] |
|
nanosized rice husk |
- |
24.75 |
- |
L, F, Te, D-R |
PFO, PSO, W-M |
medium |
[36] |
|
nanosized rice husk |
- |
76.92 |
- |
L, F, Te, D-R |
PFO, PSO, W-M |
medium |
[39] |
|
cotton plant ash |
2 |
0.64 |
- |
L, F |
PFO, PSO, W-M |
low |
[42] |
|
mustard plant ash |
- |
0.76 |
- |
L, F, Te |
PFO, PSO |
low |
[43] |
|
groundnut shell ash |
22 |
0.87 |
- |
L, F, Te |
PFO, PSO |
low |
[44] |
|
groundnut shell ash |
8 |
0.87 |
- |
L, F, Te |
PFO, PSO |
low |
[45] |
|
wheat straw ash |
37 |
1.89 |
- |
L, F, Te |
PFO, PSO |
low |
[46] |
|
functionalized Physalis peruviana biomass |
- |
233.3 |
- |
ITM |
- |
high |
[47] |
|
acid-treated peanut skin |
- |
246.7 |
- |
L, F, Te, D-R |
PFO, PSO, E |
high |
[48] |
|
apple shell |
- |
40.08 |
- |
L, F |
contact time |
medium |
[40] |
|
orange peel |
- |
22.71 |
- |
L, F |
contact time |
medium |
[40] |
|
banana peel |
- |
33.26 |
- |
L, F |
contact time |
medium |
[40] |
|
millet waste |
- |
45.45 |
- |
L, F |
contact time |
medium |
[40] |
|
coffee wastes |
- |
- |
340.0 |
L, F |
PFO, PSO |
high |
[41] |
|
N-cetylpyridinium-modified tiger nut residue |
0.03 |
79.3 |
- |
L, F, K-C |
PFO, PSO, E, W-M |
medium |
[50] |
|
H2SO4-modified wheat husks |
- |
161.1 |
- |
L, F |
Ba |
high |
[49] |
|
saw-sedge |
0.6 |
65.58 |
- |
L, F |
PFO, PSO |
medium |
[51] |
|
biomass-based MOF composite |
71 |
79.2 |
- |
L, F, Te, K-C |
PFO, PSO |
medium |
[52] |
|
biomass-based ferrospinel composite |
528 |
400 |
- |
L, F |
PFO, PSO, W-M |
high |
[53] |
Comments on the Quality of English Language
There are a lot of grammar errors that jeopardize the quality of the work done, and major revisions should be considered prior to publication.
Response to Reviewer 5 suggestion:
The manuscript was checked and corrected.

Round 2
Reviewer 1 Report
Comments and Suggestions for Authors
The authors made necessary corrections, I recommend publication.
Reviewer 4 Report
Comments and Suggestions for Authors
The correction is OK